# Understanding the role of importance weighting for deep learning

**Da Xu**
Walmart Labs
Sunnyvale, CA 94086, USA
`DaXu5180@gmail.com`

**Yuting Ye**
Division of Biostatistics
University of California, Berkeley
Berkeley, CA 94720, USA
`yeyt@berkeley.edu`

**Chuanwei Ruan** *
Instacart
San Francisco, CA 94107, USA
`Ruanchuanwei@gmail.com`

## Abstract

The recent paper by Byrd & Lipton (2019), based on empirical observations, raises a major concern on the impact of importance weighting for the over-parameterized deep learning models. They observe that as long as the model can separate the training data, the impact of importance weighting diminishes as the training proceeds. Nevertheless, there lacks a rigorous characterization of this phenomenon. In this paper, we provide formal characterizations and theoretical justifications on the role of importance weighting with respect to the implicit bias of gradient descent and margin-based learning theory. We reveal both the optimization dynamics and generalization performance under deep learning models. Our work not only explains the various novel phenomenons observed for importance weighting in deep learning, but also extends to the studies where the weights are being optimized as part of the model, which applies to a number of topics under active research.

## 1 Introduction

Importance weighting is a standard tool for estimating a quantity under a target distribution while only the samples from some source distribution is accessible. It has been drawing extensive attention in the communities of statistics and machine learning. Causal inference for deep learning investigates heavily on the propensity score weighting method that applies the off-policy optimization with counterfactual estimator (Gilotte et al., 2018; Jiang & Li, 2016), modelling with observational feedback (Schnabel et al., 2016; Xu et al., 2020) and learning from controlled intervention (Swaminathan & Joachims, 2015). The importance weighting methods are also applied to characterize distribution shifts for deep learning models (Fang et al., 2020), with modern applications in such as the domain adaptation (Azizzadenesheli et al., 2019; Lipton et al., 2018) and learning from noisy labels (Song et al., 2020). Other usages include curriculum learning (Bengio et al., 2009) and knowledge distillation (Hinton et al., 2015), where the weights characterize the model confidence on each sample.

To reduce the discrepancy between the source and target distribution for model training, a standard routine is to minimize a weighted risk (Rubinstein & Kroese, 2016). Many techniques have been developed to this end, and the common strategy is re-weighting the classes proportionally to the inverse of their frequencies (Huang et al., 2016; 2019; Wang et al., 2017). For example, Cui et al.

---

* The work was done when the author was with Walmart Labs.

(2019) proposes re-weighting by the inverse of effective number of samples. The *focal loss* (Lin et al., 2017) down-weights the well-classified examples, and the work by Li et al. (2019) suggests an improved technique which down-weights examples based on the magnitude of the gradients.

Despite the empirical successes of various re-weighting methods, it is ultimately not clear how importance weighting lays influence from the theoretical standpoint. The recent study of Byrd & Lipton (2019) observes from experiments that there is little impact of importance weights on the converged deep neural network, if the data can be separated by the model using gradient descent. They connect this phenomenon to the implicit bias of gradient descent (Soudry et al., 2018) - a novel topic that studies why over-parameterized models trained on separable data is biased toward solutions that generalize well. Implicit bias of gradient descent has been observed and studied for linear model (Soudry et al., 2018; Ji & Telgarsky, 2018b), linear neural network (Ji & Telgarsky, 2018a; Gunasekar et al., 2018), two-layer neural network with homogeneous activation (Chizat & Bach, 2020) and smooth neural networks (Nacson et al., 2019; Lyu & Li, 2019). To summarize, those work reveals that the direction of the parameters (for linear predictor) and the normalized margin (for nonlinear predictor), regardless of the initialization, respectively converge to those of a max-margin solution. The pivotal role of margin for deep learning models has been explored actively after the long journey of understanding the generalization of over-parameterized neural networks (Bartlett et al., 2017; Golowich et al., 2018; Neyshabur et al., 2018). For instance, Wei et al. (2019) studies the margin of the neural networks for separable data under weak regularization. They show that the normalized margin also converges to the max-margin solution, and provide a generalization bound for a neural network that hinges on its margin.

Although there are rich understandings for the implicit bias of gradient descent and the margin-based generalization, very few efforts are dedicated to studying how they adjust to the weighted empirical-risk minimization (ERM) setting. The established results do not directly transfer since importance weighting can change both the optimization geometry and how the generalization is measured. In this paper, we fill in the gap by showing the impact of importance weighting on the implicit bias of gradient descent as well as the generalization performance. By studying the optimization dynamics of linear models, we first reveal the effect of importance weighting on the convergence speed under linearly separable data. When the data is not linearly separable, we characterize the unique role of importance weighting on defining the intercept term upon the implicit bias. We then investigate the non-linear neural network under a weak regularization as Wei et al. (2019). We provide a novel generalization bound that reflects how importance weighting leads to the interplay between the empirical risk and a compounding term that consists of the model complexity as well as the deviation between the source target distribution. Based on our theoretical results, we discuss several exploratory developments on importance weighting that are worthy of further investigations.

- A good set of weights for learning can be inversely proportional to the hard-to-classify extent. For example, a sample that is close to (far from) the oracle decision boundary should have a large (small) weight.
- If the importance weights are jointly trained according to a weighting model, the impact of the weighting model eventually diminishes after showing strong correlation with the hard-to-classify extent such as margin.
- The usefulness of explicit regularization on weighted ERM can be studied, via their impact on the margin, on balancing the empirical loss and the distribution divergence.

In summary, our contribution are three folds.

- We characterize the impact of importance weighting on the implicit bias of gradient descent.
- We find a generalization bound that hinges on the importance weights. For finite-step training, the role of importance weighting on the generalization bound is reflected in how the margin is affected, and how it balances the source and target distribution.
- We propose several exploratory topics for importance weighting that worth further investigating from both the application and theoretical perspective.

The rest of the paper is organized as follows. In Section 2, we introduce the background, preliminary results and the experimental setup. In Section 3 and 4, we demonstrate the influence of the importance weighting for linear and non-linear models in terms of the implicit bias of gradient descent and the generalization performance. We then discuss the extended investigations in Section 5.

## 2 PRELIMINARIES

We use bold-font letters for vectors and matrices, uppercase letters for random variables and distributions, and $\|\cdot\|$ to denote $\ell_2$ norm when no confusion arises. We denote the training data by $\mathcal{D} = \{w_i, \mathbf{x}_i, y_i\}_{i=1}^n$ where $\mathbf{x}_i \in \mathcal{X}$ denotes the features, $y_i$ is binary or categorical, and the importance weight is bounded such that: $w_i \in [1/M, M]$ for some $M > 1$. We mention that the importance weights are often defined with respect to the *source distribution* $P_s$ from which the training data is drawn, and the *target distribution* $P_t$. We do not make this assumption here because importance weighting is often applied for more general purposes. Therefore, $w_i$ can be defined arbitrarily.

We use $f(\boldsymbol{\theta}, \mathbf{x})$ to denote the predictor and define $\mathcal{F} = \{f(\boldsymbol{\theta}, \cdot) \,|\, \theta \in \boldsymbol{\Theta} \subset \mathbb{R}^d\}$. For the sake of notation, we focus on the binary setting: $y_i \in \{-1, +1\}$ with $f(\boldsymbol{\theta}, \mathbf{x}) \in \mathbb{R}$. However, it will become clear later that our results can be easily extended to the multi-class setting. Consider the weighted *empirical risk minimization* (ERM) task with the risk given by $L(\boldsymbol{\theta}; \mathbf{w}) = 1/n \sum_{i=1}^n w_i \ell\big(y_i f(\boldsymbol{\theta}, \mathbf{x}_i)\big)$ for some non-negative loss function $\ell(\cdot)$. The weight-agnostic counterpart is denoted by: $L(\boldsymbol{\theta}) = 1/n \sum_{i=1}^n \ell(y_i f(\boldsymbol{\theta}, \mathbf{x}_i))$. We focus particularly on the exponential loss $\ell(u) = \exp(-u)$ and log loss $\ell(u) = \log(1 + \exp(-u))$. For the multi-class problem where $y_i \in [k]$, we extend our setup using the softmax function where the logits are now given by $\{\boldsymbol{f}_j(\boldsymbol{\theta}, \mathbf{x})\}_{j=1}^k$. For optimization, we consider using gradient descent to minimize the total loss: $\boldsymbol{\theta}^{(t+1)}(\mathbf{w}) = \boldsymbol{\theta}^{(t)}(\mathbf{w}) - \eta_t \nabla L(\boldsymbol{\theta}; \mathbf{w})\big|_{\boldsymbol{\theta} = \boldsymbol{\theta}^{(t)}(\mathbf{w})}$, where the learning rate $\eta_t$ can be constant or step-dependent.

**From parameter norm divergence to support vectors.**

Suppose $\mathcal{D}$ is separated by $f(\boldsymbol{\theta}^{(t)}, \mathbf{x})$ after some point during training. The key factor that contributes to the implicit bias for both linear and non-linear predictor under a *weak regularization* [1] is that the norm of the parameters diverges after separation, i.e. $\lim_{t\to\infty} \|\boldsymbol{\theta}^{(t)}\|_2 = \infty$, as a consequence of using gradient descent. Now we examine $\|\boldsymbol{\theta}^{(t)}(\mathbf{w})\|_2$. The heuristic is that if $\ell(\cdot)$ is exponential-like, multiplying by $w_i$ only changes its tail property up to a constant while the asymptotic behavior is not affected. In particular, the necessary conditions for norm divergence under gradient descent can be summarized by:

- **C1**. The loss function $\ell(\cdot)$ has a exponential tail behavior (that we formalize in Appendix A.1) such that $\lim_{u\to\infty} \ell(-u) = \lim_{u\to\infty} \nabla\ell(-u) = 0$;
- **C2**. The predictor $f(\boldsymbol{\theta}, \mathbf{x})$ is $\alpha$-homogeneous such that $f(c \cdot \boldsymbol{\theta}, \mathbf{x}) = c^\alpha f(\boldsymbol{\theta}, \mathbf{x}), \forall c > 0$.

In addition, we need certain regularities from $f(\boldsymbol{\theta}, \mathbf{x})$ to ensure the existence of critical points and the convergence of gradient descent:

- **C3**. for any $\mathbf{x} \in \mathcal{X}$, $f(\cdot, \mathbf{x})$ is $\beta$-smooth and $l$-Lipschitz on $\mathbb{R}^d$.

**C1** can be satisfied by the exponential loss, log loss and cross entropy loss under the multi-class setting. For standard deep learning models such as multilayer perceptron (MLP), **C2** implies that the activation functions are homogeneous such as ReLU and LeakyReLU, and bias terms are disallowed. **C3** is a common technical assumptions whose practical implications are discussed in Appendix A.1. Among the three necessary conditions, importance weighting only affects **C1** up to a constant, so its impact on the norm divergence diminishes in the asymptotic regime. The formal statement is provided as below.

**Claim 1.** *There exists a constant learning rate for gradient descent, such that for any* $\mathbf{w} \in [1/M, M]^n$, *with a weak regularization,* $\lim_{t\to\infty} \big\|\boldsymbol{\theta}^{(t)}(\mathbf{w})\big\| = \infty$ *under **C1-C3**.*

Compared with the previous work, we extend the norm divergence result not only to weighted ERM but a more general setting where a weak regularization is considered. We defer the proof to Appendix A.1. A direct consequence of parameter norm divergence is that both the risk and the gradient are dominated by the terms with the smallest margin, i.e. $\arg\min_i y_i f(\boldsymbol{\theta}, \mathbf{x}_i)$, which are also referred to as the "support vectors". To make sense of this point, notice that both the risk and the gradient have the form of: $\sum_i C_i \exp\big(-y_i f(\boldsymbol{\theta}, \mathbf{x}_i)\big)$, where $C_i$ are low-order terms. Since $f(\boldsymbol{\theta}, \mathbf{x}_i) = \|\boldsymbol{\theta}\|_2^\alpha f(\boldsymbol{\theta}/\|\boldsymbol{\theta}\|_2, \mathbf{x}_i)$ due to the homogeneous assumption in **C2**, it holds that:

---

[1]The regularized loss is given by $L_\lambda(\boldsymbol{\theta}; \mathbf{w}) = L(\boldsymbol{\theta}; \mathbf{w}) + \lambda\|\boldsymbol{\theta}\|^r$ for a fixed $r > 0$. The weak regularization refers to the case where $\lambda \to 0$.

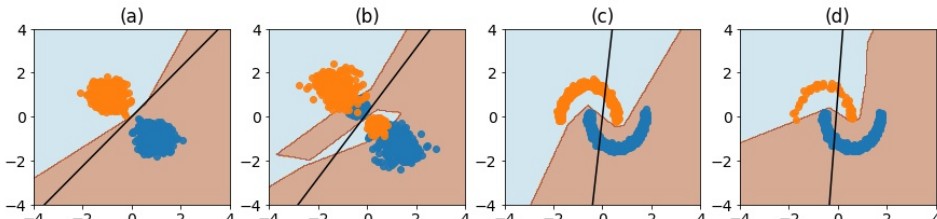

Figure 1: **(a)**. Linearly separable data; **(b)**. Non-separable data; **(c)**: Balanced moon-shaped non-linear separable data; **(d)**. Unbalance moon-shaped data after down-sampling both classes (20% for the blue class, and 80% for the orange class). We use solid line to denote the separating hyperplane of the trained linear model and shades to represent the decision boundary of trained nonlinear model.

$\lim_{t \to \infty} \exp\big(-y_i f(\boldsymbol{\theta}^{(t)}(\mathbf{w}), \mathbf{x}_i)\big) \to 0$. Therefore, the decision boundaries may share certain characteristics with the support vector machine (SVM) since they rely on the same support vectors. As a matter of fact, the current understandings on the implicit bias of gradient descent are mostly established on the connection with hard-margin SVM:

$$\min_{\boldsymbol{\theta} \in \mathbb{R}^d} \|\boldsymbol{\theta}\|_2 \quad \text{s.t.} \quad y_i f(\boldsymbol{\theta}, \mathbf{x}_i) \geq 1 \quad \forall i = 1, 2, \ldots, n, \tag{1}$$

whose optimization path coincides with the max-margin problem: $\max_{\|\boldsymbol{\theta}\|_2 \leq 1} \min_{i=1,\ldots,n} y_i f(\boldsymbol{\theta}, \mathbf{x}_i)$, as shown by Nacson et al. (2019). Define $\gamma(\boldsymbol{\theta}) := \min_i y_i f(\boldsymbol{\theta}, \mathbf{x}_i)$. We use $\boldsymbol{\theta}^*$ to denote the optimal solution and $\gamma^* = \gamma(\boldsymbol{\theta}^*) := \min_i y_i f(\boldsymbol{\theta}^*, \mathbf{x}_i)$ to denote the corresponding margin.

**Implicit bias of gradient descent.**

We start by considering the weight-agnostic setting. When $\mathcal{D}$ is linear separable, it is reasonable to conjecture that the separating hyperplane under a linear $f(\boldsymbol{\theta}, \cdot)$ overlaps with the solution of hard-margin SVM. Soudry et al. (2018) and Ji & Telgarsky (2018b) first show that $\|\boldsymbol{\theta}^{(t)}\|$ converges in direction to $\boldsymbol{\theta}^*$, i.e. $\lim_{t \to \infty} \boldsymbol{\theta}^{(t)}/\|\boldsymbol{\theta}^{(t)}\|_2 = \boldsymbol{\theta}^*$. For nonlinear predictors, however, the parameter direction is less meaningful. Instead, it has been pointed out that neural networks often achieve perfect separation of the training data (Zhang et al., 2016). Therefore, we are more interested in the margin whose pivoting role for the generalization of neural networks is studied extensively (Neyshabur et al., 2017; Bartlett et al., 2017; Golowich et al., 2018). Specifically, it has been show in Nacson et al. (2019) and Lyu & Li (2019) that the *normalized margin*, defined by $\tilde{\gamma}(\boldsymbol{\theta}^{(t)}) := \gamma\big(\boldsymbol{\theta}^{(t)}/\|\boldsymbol{\theta}^{(t)}\|_2\big)$, converges to the maximum margin $\gamma^*$ without regularization.

It becomes clear at this point that to understand the role of importance weighting for deep learning, we must characterize the impact of weights on the implicit bias since they reveal the optimization geometry and generalization performance. Formally, we address the following critical questions.

- **Q1**. Does importance weighting modify the convergence results (convergence in direction for linear predictor and in normalized margin for nonlinear predictor)?

- If the convergence results remain unchanged, then:

    - **Q2**. in what way is importance weighting affecting the optimization process;

    - **Q3**. how does importance weighting influence the generalization from the source distribution to the target distribution?

**Experiment setup.**

Throughout this paper, we use the regular regression model as linear predictor. The nonlinear predictor is a two-layer MLP with five hidden units and ReLU as the activation function. All the models are trained with gradient descent using 0.1 as learning rate. We use the exponential loss and the standard normal initialization. The generated datasets for our illustrative experiments are shown in Figure 1, which correspond to the different settings of our major topics.

## 3 IMPORTANCE WEIGHTING FOR LINEAR PREDICTOR

We begin with the linear predictors which allows more refined analysis on the gradient dynamics. Without loss of generality, we assume using the exponential loss. Also, we do not consider the weak regularization here since its practical impact on linear model is trivial when $\lambda \to 0$ (Rosset et al., 2004a;b), but it is not the case for nonlinear predictors. One sophistication with linear predictor is that the data may not be perfectly separated, as opposed to the nonlinear case where neural networks can in theory separate any non-degenerate data. With this kept in mind, we first assume $\mathcal{D}$ is linear separable and characterize the new convergence result in the following proposition.

**Proposition 1.** *With a constant learning rate $\eta_t \lesssim \beta^{-1}$, we consider normalizing the weights $\mathbf{w} \in [\frac{1}{M}, M]^n$ such that $\sum_i \mathbf{w}_i = 1$ without loss of generality, it holds that:*

$$\left| \frac{\boldsymbol{\theta}^{(t)}(\mathbf{w})}{\|\boldsymbol{\theta}^{(t)}(\mathbf{w})\|_2} - \boldsymbol{\theta}^* \right| \lesssim \frac{\log n + D_{KL}(\boldsymbol{p}^* \| \mathbf{w}) + M}{\log t \cdot \gamma^*}, \tag{2}$$

*where $\boldsymbol{p}^* = [p_1^*, \dots, p_n^*]$ characterizes the dual optimal for the hard-margin SVM such that $\boldsymbol{\theta}^* = \sum_{i=1}^n y_i \mathbf{x}_i \cdot p_i^*$ and satisfies: $p_i^* \geq 0$ and $\sum_{i=1}^n p_i^* = 1$. Here, $D_{KL}$ is the Kullback-Leibler divergence.*

We leave the proof to Appendix A.2. We find that importance weighting does not change the convergence result as well as the $1/\log t$ convergence rate. However, it does affect the convergence speed under the finite-step optimization. In particular, we show that the extra constant term induced by importance weighting is given by the KL-divergence between the (normalized) weights and the dual optimal of the hard-margin SVM, where samples with smaller margins usually have larger values. Therefore, importance weighting may accelerate gradient descent in finite-step optimization by matching weights with the inverse margin. As we show in Figure 2a and 2b, this type of "inverse-margin weighted" design is able to accelerate the convergence and bring better performance under finite-step optimization.

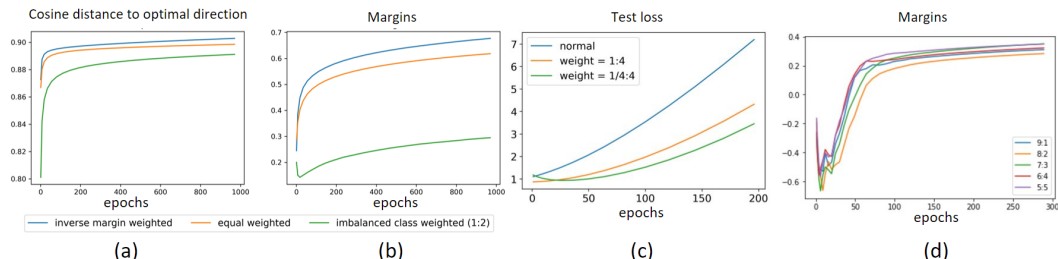

(a)         (b)         (c)         (d)

Figure 2: **(a)**: Epoch-wise training performances measured by the angle between the decision boundary (at that epoch) and the max-margin solution, using linear predictor on the linear separable data of Figure 1a; **(b)**: Epoch-wise training performances measured by the average margin in the same setting as (a); **(c)**. The generalization error on testing data (the remaining 80% of the orange class and 20% of the blue class that are not part of the down-sampling in Figure 1d) when the nonlinear model is trained under different class weights, as the training progresses; **(d)**. The average margin for the nonlinear model on the non-linearly separable training data shown in Figure 1c, under different class weights, as the training progresses.

When $\mathcal{D}$ is not linearly separable, the key insight is that we can always partition $\mathcal{D}$ into $\mathcal{D}_{\text{sep}} \cup \mathcal{D}_{\text{non-sep}}$, where $\mathcal{D}_{\text{sep}}$ is the maximal linear separable subset defined in Ji & Telgarsky (2018b). Let $\Pi_{\text{non-sep}}$ be the (orthogonal) projection onto the subspace $S$ spanned by the $\mathbf{x}_i$'s in $\mathcal{D}_{\text{non-sep}}$, and let $\Pi_{\text{sep}}$ be the projection onto the orthogonal complement $S^{\perp}$. The partition allows us to study the two projected parts independently since by the construction, we have $\boldsymbol{\theta}^{(t)}(\mathbf{w}) = \Pi_{\text{non-sep}}\boldsymbol{\theta}^{(t)}(\mathbf{w}) + \Pi_{\text{sep}}\boldsymbol{\theta}^{(t)}(\mathbf{w})$. It is intuitive that the optimization path of $\Pi_{\text{sep}}\boldsymbol{\theta}^{(t)}(\mathbf{w})$ behaves similarly to the linear separable case as in Proposition 1, so we can focus on the properties of $\Pi_{\text{non-sep}}\boldsymbol{\theta}^{(t)}(\mathbf{w})$, which we summarize in the follow proposition.

**Proposition 2** (Informal). *Let $L_{non\text{-}sep}(\boldsymbol{\theta}, \mathbf{w})$ be the weighted risk defined on the non-separable subset, then with the constant learning rate:*

- $\tilde{\boldsymbol{\theta}}(\mathbf{w}) = \arg\min_{\boldsymbol{\theta}} L_{non\text{-}sep}(\boldsymbol{\theta}, \mathbf{w})$ *is uniquely defined and $\left\|\tilde{\boldsymbol{\theta}}(\mathbf{w})\right\|_2 = \mathcal{O}(1)$;*

- $\left| \Pi_{non\text{-}sep} \boldsymbol{\theta}^{(t)}(\mathbf{w}) - \tilde{\boldsymbol{\theta}}(\mathbf{w}) \right| \lesssim \dfrac{C\left( \left\| \tilde{\boldsymbol{\theta}}(\mathbf{w}) \right\|_2 \right) + \log^2 t / \gamma_{sep}}{t}$, *where $\gamma_{sep}$ is the maximum margin on $\mathcal{D}_{sep}$ and $C\left( \left\| \tilde{\boldsymbol{\theta}}(\mathbf{w}) \right) \right\|_2 \right) = \mathcal{O}(1)$.*

The formal statement, which involves how $\mathcal{D}_{\text{sep}}$ is defined, is deferred to Appendix A.2 together with the proof. Proposition 2 informs that importance weighting uniquely defines the solution $\tilde{\boldsymbol{\theta}}(\mathbf{w})$ on the non-separable subset of the data, to which $\Pi_{\text{non-sep}} \boldsymbol{\theta}^{(t)}(\mathbf{w})$ converges. Hence, we expect $\lim_{t \to \infty} \boldsymbol{\theta}^{(t)}(\mathbf{w}) = \tilde{\boldsymbol{\theta}}(\mathbf{w}) + \boldsymbol{\theta}^*_{\text{sep}}$, where $\boldsymbol{\theta}^*_{\text{sep}}$ is the solution on the separable subset $\mathcal{D}_{\text{sep}}$ and thus its direction does not depend on $\mathbf{w}$ as implied by Proposition 1. We can therefore think of $\tilde{\boldsymbol{\theta}}(\mathbf{w})$ as the intercept term where the weight controls how the intercept shifts on the subspace of the non-separable data. We also illustrate this finding in Figure 3. By far, we provide an in-depth understanding and our theoretical results fully explain the observations made in Byrd & Lipton (2019) on how importance weighting affects the implicit bias of gradient descent using linear predictors.

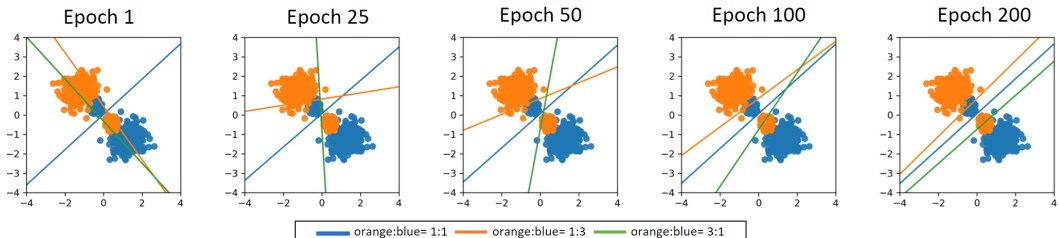

Figure 3: The role of importance weighting on defining the intercept term in addition to the implicit bias for the linearly separable case, where the hyperplane shifts in the non-separable subspace depending on the class weights.

## 4 IMPORTANCE WEIGHTING FOR NONLINEAR PREDICTOR

Now we investigate the influence of importance weighting on non-linear predictors, e.g, the neural network. Here we are more interested in the regularized setting:

$$\min_{\boldsymbol{\theta}} L_\lambda(\boldsymbol{\theta}; \mathbf{w}) := L(\boldsymbol{\theta}, \mathbf{w}) + \lambda \|\boldsymbol{\theta}\|^r, \tag{3}$$

where $r > 0$ is fixed, $\lambda$ is the regularization coefficient. We use the notation: $\boldsymbol{\theta}_\lambda(\mathbf{w}) \in \arg\min L_\lambda(\boldsymbol{\theta}, \mathbf{w})$. Recall that $\gamma^* := \max_{\|\boldsymbol{\theta}\| \leq 1} \min_i y_i f(\boldsymbol{\theta}, \mathbf{x}_i)$. Unlike the linear case, characterizing the gradient dynamics for nonlinear predictor is often insurmountable. Therefore, we mainly consider the asymptotic regime or the regime with sufficiently large $t$. We omit the superscript in $\boldsymbol{\theta}^{(t)}$ when there is no confusion. The only assumption we need to make is that:

**A1**. the data is separated by $f$ at some point during gradient descent, i.e. $\exists t > 0$ s.t. $y_i f(\boldsymbol{\theta}^{(t)}, \mathbf{x}_i) > 0, \forall i = 1, \ldots, n$. In addition, $y_i f(\boldsymbol{\theta}^*, \mathbf{x}_i) \geq \gamma^* > 0$ for each $i$.

In Section 4.1, we show that by solving the equation 3 with an infinitesimal (weak) regularizer, gradient descent leads to the optimal margin $\gamma^*$, regardless of the choice of the importance weights. In Section 4.2, we show that the the importance weighting affects the generalization bound via a multiplication factor as well as the margin in the finite-sample scenario.

### 4.1 MARGIN IS INVARIANT TO IMPORTANCE WEIGHTING UNDER WEAK REGULARIZATION

We show that for any bounded $\mathbf{w}$, $\tilde{\gamma}(\boldsymbol{\theta}_\lambda(\mathbf{w})) := \gamma(\boldsymbol{\theta}_\lambda(\mathbf{w}) / \|\boldsymbol{\theta}_\lambda(\mathbf{w})\|)$ converges to $\gamma^*$ as $\lambda$ decreases to zero. In practice, however, we might not obtain $\boldsymbol{\theta}_\lambda(\mathbf{w})$ in limited time. It is shown that as long as equation 3 is close enough to its optimum, the normalized margin of the associated $\boldsymbol{\theta}'(\mathbf{w})$ (under finite-step optimization) is lower bounded by $\gamma^*$ multiplied by a non-trivial factor. Formally,

**Proposition 3.** *Suppose C1-C3, A1 hold. For any $\mathbf{w} \in [1/M, M]^n$, it follows that*

- *(Asymptotic)* $\lim_{\lambda \to 0} \tilde{\gamma}(\boldsymbol{\theta}_\lambda(\mathbf{w})) \to \gamma^*$.

- **(Finite steps)** *There exists a $\lambda := \lambda(r, \alpha, \gamma^*, \mathbf{w}, c)$ such that for $\boldsymbol{\theta}'(\mathbf{w})$ with $L_\lambda(\boldsymbol{\theta}'(\mathbf{w}); \mathbf{w}) \leq \tau L_\lambda(\boldsymbol{\theta}_\lambda(\mathbf{w}); \mathbf{w})$ and $\tau \leq 2$, the associated normalized margin $\tilde{\gamma}(\boldsymbol{\theta}'(\mathbf{w}))$ satisfies $\tilde{\gamma}(\boldsymbol{\theta}'(\mathbf{w})) \geq c \cdot \frac{\gamma^*}{\tau^{\alpha/r}}$, where $\frac{1}{10} \leq c < 1$.*

This result is adapted from Wei et al. (2019), which relies on Claim 1. The proof is relegated to Appendix A.4.1. We see that importance weighting does not affect the asymptotic margin when $\lambda$ is sufficiently small. To get the intuition, note that when $\|\boldsymbol{\theta}_\lambda(\mathbf{w})\|$ is large enough and $\lambda$ is small enough to be ignored, $L_\lambda(\boldsymbol{\theta}_\lambda(\mathbf{w}), \mathbf{w}) \approx \exp\left(-\|\boldsymbol{\theta}_\lambda(\mathbf{w})\|^\alpha \gamma_\lambda\right)$, which favors a large margin. In addition, even if $L_\lambda(\boldsymbol{\theta}'(\mathbf{w}), \mathbf{w})$ has not yet converged but close enough to its optimum, the corresponding normalized margin has a reasonable lower bound. We point out that this result does not rely on the choice of $\lambda$. The assumption $L_\lambda(\boldsymbol{\theta}'(\mathbf{w}); \mathbf{w}) \leq \tau L_\lambda(\boldsymbol{\theta}_\lambda(\mathbf{w}); \mathbf{w})$ has already accounted for the major influence of importance weighting in terms of the optimization. That is, with a "good" set of importance weights, we can achieve this criteria (by approaching global optimum) faster. We leave detailed discussions to Section 5. Figure 2d also demonstrates that the choice of the importance weights has a significant influence on the convergence speed for the non-linear predictor.

## 4.2 IMPORTANCE WEIGHTING AFFECTS THE GENERALIZATION BOUND

Proposition 3 conjectures on the behavior of the margin corresponding to the optimum of $L_\lambda(\boldsymbol{\theta}; \mathbf{w})$, which does not rely on the sample size. To bridge the connection between importance weighting and the behavior of $f(\boldsymbol{\theta}, \cdot)$ in the finite-sample setting, we investigate the generalization bound of $f$ when the training sample distribution deviates from the testing sample distribution.

Let $P_s$ be the source distribution and $P_t$ be the target distribution with the corresponding densities $p_s(\cdot)$ and $p_t(\cdot)$. Assume that $P_s$ and $P_t$ have the same support. We consider the Pearson $\chi^2$-divergence to measure the difference between $P_s$ and $P_t$, i.e., $D_{\chi^2}(P_t \| P_t) = \int \left[(dP_s/dP_t)^2 - 1\right] dP_s$. The training covariates $\mathbf{x}_1, \ldots, \mathbf{x}_n$ are generated from $P_s$, and the testing covariates are generated from $P_t$. Denote by $p_{\text{train}}$ and $p_{\text{test}}$ the joint distribution of $(\mathbf{x}, y)$ for the training data and the testing data, respectively.

We minimize equation 3 over the $H$-layer feedforward neural network given by $f^{\text{NN}}(\boldsymbol{\theta}, \mathbf{x}) := W_H \sigma(W_{H-1}\sigma(\cdots \sigma(W_1 \mathbf{x}) \cdots))$, where $\boldsymbol{\theta} = [W_1, \cdots, W_H]$ are the parameter matrices and $\sigma(\cdot)$ is the element-wise activation function such as ReLU. Denote by $\eta(\mathbf{x}) = p_t(\mathbf{x})/p_s(\mathbf{x})$. We show that the generalization performance is affected by importance weighting via the interplay between the empirical risk that hinges on $\boldsymbol{\eta}$, as well as a term that depends on the model complexity and the deviation of the target distribution from the source distribution.

**Theorem 1** (1). *Assume $\sigma$ is 1-Lipschitz and 1-positive homogeneous. Then with probability at least $1 - \delta$, we have*

$$\mathbb{P}_{(\mathbf{x}, y) \sim p_{\text{test}}}\left(y f^{NN}(\boldsymbol{\theta}(\mathbf{w}), \mathbf{x}) \leq 0\right) \leq$$

$$\underbrace{\frac{1}{n} \sum_{i=1}^{n} \eta(\mathbf{x}_i) \mathbf{I}\left(y_i f^{NN}(\boldsymbol{\theta}(\mathbf{w})/\|\boldsymbol{\theta}(\mathbf{w})\|, \mathbf{x}_i) < \gamma\right)}_{(I)} + \underbrace{\frac{C \cdot \sqrt{D_{\chi^2}(P_t \| P_s) + 1}}{\gamma \cdot H^{(H-1)/2}\sqrt{n}}}_{(II)} + \epsilon(\gamma, n, \delta),$$

*where (I) is the empirical risk, (II) reflects the compounding effect of the model complexity of the class of $H$-layer neural networks and the deviation between target distribution and source distribution, $\epsilon(\gamma, n, \delta) = \sqrt{\frac{\log \log_2 \frac{4C}{\gamma}}{n}} + \sqrt{\frac{\log(1/\delta)}{n}}$ is a small quantity compared to (I) and (II). Here, $C := \sup_{\mathbf{x}} \|\mathbf{x}\|$ and $\gamma$ can take any positive value.*

The proof is deferred to Appendix A.4.2. Compared to Wei et al. (2019), the empirical risk (I) hinges on $\boldsymbol{\eta}$ and there is an additional multiplier factor $\sqrt{D_{\chi^2}(P_t \| P_s) + 1}$ on (II). In the two discussions below, we argue that the role of importance weighting on the generalization bound in Theorem 1 is not only reflected in how the margin is affected, but also how it balances source and target distribution:

**1.** Suppose $\boldsymbol{\theta}(\mathbf{w})$ enables $f^{\text{NN}}$ to separate the data. Let $\gamma_{\boldsymbol{\theta}(\mathbf{w})} := \min_i y_i f^{\text{NN}}(\boldsymbol{\theta}(\mathbf{w})/\|\boldsymbol{\theta}(\mathbf{w})\|, \mathbf{x}_i)$. In the generalization bound of Theorem 1, if we let $\gamma = \gamma_{\boldsymbol{\theta}(\mathbf{w})}$, then (I) vanishes and only (II) remains. In this case, the importance weights affects the generalization bound via $\gamma_{\boldsymbol{\theta}(\mathbf{w})}$ in finite steps as

discussed in Section 4.1. That is, within finite training steps, a good set of weights $\mathbf{w}$ can approach closer to $\gamma_{\boldsymbol{\theta}(\mathbf{w})}$ than a bad set, and thus giving a better generalization performance. Also note that Theorem 1 holds for the non-separable cases as well.

**2.** We point out that (II) is a strictly decreasing function, while (I) is a non-decreasing step function with respect to $\gamma$. Therefore, there must exists a trade-off $\gamma$ that minimizes the sum of (I) and (II), which is usually attained at some $\gamma > \gamma_{\boldsymbol{\theta}(\mathbf{w})}$. When $\gamma$ grows, certain samples will activate $\mathbf{I}(y_i f^{\text{NN}}(\boldsymbol{\theta}(\mathbf{w})/\|\boldsymbol{\theta}(\mathbf{w})\|, \mathbf{x}_i) < \gamma)$ and inflate (I). The hope is that an initially activated sample (indicator term) in (I) corresponds to a small $\eta(\mathbf{x}_i)$, while one with a large $\eta(\mathbf{x}_{i'})$ has a large value of $y_{i'} f^{\text{NN}}(\boldsymbol{\theta}(\mathbf{w})/\|\boldsymbol{\theta}(\mathbf{w})\|, \mathbf{x}_{i'})$ and thus will be activated later. This can be achieved by aligning $\mathbf{w}$ with $\boldsymbol{\eta}$ because a large weight on sample $i$ forces the decision boundary to drift away from this data point and gives a larger value of $y_i f^{\text{NN}}(\boldsymbol{\theta}(\mathbf{w})/\|\boldsymbol{\theta}(\mathbf{w})\|, \mathbf{x}_i)$. Therefore, the generalization bound with $\mathbf{w}$ aligning with $\boldsymbol{\eta}$ can be smaller than that with $\mathbf{w}$ deviating from $\boldsymbol{\eta}$.

The empirical results in Figure 2c provides the numerical evidence that reflects the strong effects of importance weighting on the generalization behavior.

## 5 EXTENSION

**What makes a good set of weights for learning?**

We show in both Section 3 and 4 that importance weighting can affect how fast the classifier separates the data and converges to the max-margin solution. We also justify how the small-margin support vectors, who can think of as the hard-to-classify data points, are of significant importance. Imagine that we have access to an oracle that outputs the distance of each sample to the max-margin decision boundary. It is intuitive that by putting more weights on the small-margin samples, we "inform" gradient descent of their importance from the beginning and therefore accelerates the optimization. We also provide a rigorous result for linear predictor in Proposition 1. Our high-level intuition justifies a number of methodologies where people use various methods to measure the hardness of classifying a sample and use that as the weight, explicitly or implicitly. Examples include the curriculum learning (Bengio et al., 2009), mentor net (Jiang et al., 2018), co-teaching (Han et al., 2018) and knowledge distillation (Li et al., 2017; Hinton et al., 2015), where auxiliary models are employed (replacing the oracle) to represent the hardness of each data point.

**The effect of jointly optimizing a weighting model**

It is not unusual that the importance weights, when depending on another model, is jointly trained with the classifier to achieve an better overall performance, such as the counterfactual modelling (Schnabel et al., 2016; Xu et al., 2020) and learning from noisy labels (Song et al., 2020). For the illustration purpose, we consider the following setup:

$$\underset{\boldsymbol{\psi}, \boldsymbol{\theta}}{\text{minimize}} \frac{1}{n} \sum_{i=1}^{n} g(\boldsymbol{\psi}, \mathbf{x}_i) \cdot \ell\big(y_i f(\boldsymbol{\theta}, \mathbf{x}_i)\big), \quad \text{s.t.} \quad \frac{1}{M} < g(\boldsymbol{\psi}, \mathbf{x}_i) < M, \tag{4}$$

where $g(\boldsymbol{\psi}, \mathbf{x}_i)$ is the weighting model. By our main results, it is not difficult to conjecture that if the data is separable by $f$, the convergence of $f$ to the max-margin solution will still hold and the weighting model $g(\boldsymbol{\psi}, \mathbf{x}_i)$ will concentrate to a constant for all $i = 1, \dots, n$. This is because the general convergence results are agnostic to the weights, so the weighting model will eventually be nullified. Also, during the beginning phase of training, the learned weights may correlate negatively to the margin (as it helps to speed up the convergence), and the correlation will diminish eventually as the weights converge to the same constant. The above conjectures are supported by the empirical evidence that we discuss in Figure 4. Therefore, jointly optimizing the weighting model may not change the convergence result but the speed of convergence is affected.

**Interaction with explicit regularizations**

Deep learning models are often trained with explicit regularization. To see how they interact with importance weighting, we first check weather they alter the norm divergence in Claim 1. It is obvious that both the early stopping and strong regularization on $\|\boldsymbol{\theta}\|$ prohibits the norm divergence, so $f(\boldsymbol{\theta}, \cdot)$ will not achieve the max-margin solution or even separate the training data. In such cases, as it has been observed by Byrd & Lipton (2019), the impact of importance weighting on $\boldsymbol{\theta}_\lambda(\mathbf{w})$ and $\tilde{\gamma}(\boldsymbol{\theta}_\lambda(\mathbf{w}))$ will be significant. However, this may not help generalization according to our arguments

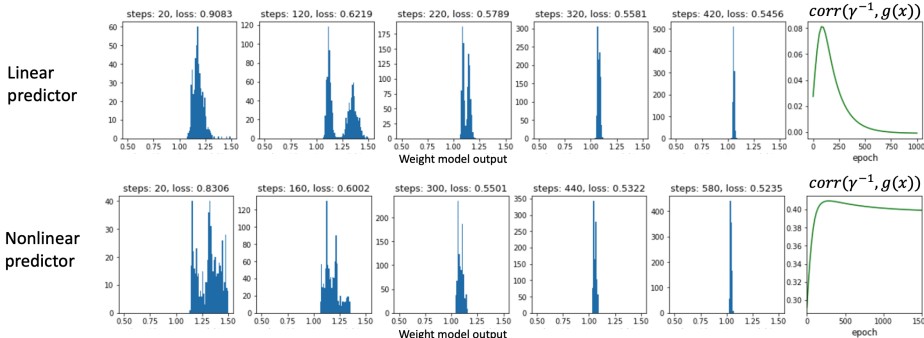

Figure 4: The left-five figures show that the distribution of the learned weights concentrates to a constant as the training progresses. The rightmost figure indicates the correlation pattern between margin and the learned weights: the correlation increases rapidly in the beginning, and then slowly decreases to zero (the process is much slower for nonlinear predictor so we only show the first part). Here, $g(\mathbf{x}_i) = \sigma(\boldsymbol{\psi}^\intercal \mathbf{x}_i + b) + 1$, where $\sigma(\cdot)$ is the sigmoid function, the constant one is added to avoid numerical issues.

in Section 4.2, since the margins will be altered as well. Indeed, Zhang et al. (2016) shows that explicit regularizations may not lead to better generalization for neural networks. For the weighted ERM, Theorem 1 provides a powerful tool to characterize the trade-off induced by explicit regularizations via the margin size. Dropout, as an counter example, does not prohibit norm divergence and may not interfere with our main conclusions.

## 6 DISCUSSION

In this paper, we study the impact of importance weighting on the implicit bias of gradient descent as well as the generalization performance. Based on our theoretical findings, we propose the following future directions that are worth investigating from both the application and theoretical perspective: 1) Is there an optimal way to construct importance weights using such as the oracle margin? 2) How to correctly understand and utilize the role of a jointly-trained weighting model? 3) What is the combined effect of importance weighting and explicit regularizations for deep learning models?

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

# A    APPENDIX

We provide the omitted discussions, proofs, and extra numerical results in the appendix.

## A.1    SUPPLEMENTARY MATERIAL FOR SECTION 2

We discuss the exponential-tail behavior for loss functions, the piratical implication of condition **C3** and the proof of Claim 1.

### A.1.1    LOSS FUNCTION WITH EXPONENTIAL-TAIL BEHAVIOR

Having a exponential decay on the tail of the loss function is essential for realizing the implicit bias of gradient descent, since we need $\ell(u)$ behave like $\exp(-u)$ as $u \to \infty$. Soudry et al. (2018) first propose the notion of *tight exponential tail*, where the negative loss derivative $-\ell'(u)$ behave like:

$$-\ell'(u) \lesssim \big(1 + \exp(-c_1 u)\big)e^{-u} \text{ and } -\ell'(u) \gtrsim \big(1 - \exp(-c_2 u)\big)e^{-u},$$

for sufficiently large $u$, where $c_1$ and $c_2$ are positive constants. There is also a smoothness assumption on $\ell(\cdot)$. It is obvious that under this definition, the tail behavior of the loss function is constraint from both sides by exponential-type functions.

There is a more general (and perhaps more direct) definition of exponential-tail loss function Lyu & Li (2019), where $\ell(u) = \exp(-f(u))$, such that:

- $f$ is smooth and $f'(u) \geq 0, \forall u$;
- there exists $c > 0$ such that $f'(u)u$ is non-decreasing for $u > c$ and $f'(u)u \to \infty$ as $u \to \infty$.

It is easy to verify that the exponential loss, log loss and cross-entropy loss satisfy both definitions. Since our focus is not to study the implicit bias of gradient descent, it suffice to work with the above loss functions.

### A.1.2    PRACTICAL IMPLICATIONS OF CONDITION **C3**

**C3** asserts the Lipschitz and smoothness properties. The Lipschitz condition is rather mild assumption for neural networks, and several recent paper are dedicated to obtaining the Lipschitz constant of certain deep learning models (Fazlyab et al., 2019; Virmaux & Scaman, 2018).

The $\beta$-smooth condition, on the other hand, is more technical-driven such that we can analyze the gradient descent. In practice, neural networks with ReLU activation do not satisfy the smoothness condition. However, there are smooth homogeneous activation functions, such as the quadratic activation $\sigma(x) = x^2$ and higher-order ReLU activation $\sigma(x) = \text{ReLU}(x)^c$ for $c > 2$. Still, in our experiments, we use ReLU as the activation function for its convenience.

### A.1.3    PROOF FOR CLAIM 1

Soudry et al. (2018) and Ji & Telgarsky (2018b) show norm divergence for linear predictors, and the follow-up work by Ji & Telgarsky (2018a); Gunasekar et al. (2018) extend the result to linear neural networks. For nonlinear predictors such as multi-layer neural network with homogeneous activation, Nacson et al. (2019) and Lyu & Li (2019) prove the norm divergence for gradient descent in the absence of explicit regularization. Rosset et al. (2004a) and Wei et al. (2019) considers the weak regularization for linear and nonlinear predictors, however, they only study the property of the critical points instead of the gradient descent sequence.

*Proof.* We first state a technical lemma that characterizes the dynamics of gradient descent.

**Lemma A.1** (Theorem E.10 of Lyu & Li (2019)). *Under the conditions that:*

- *$\ell(\cdot)$ is given by the exponential loss, and $\ell \circ f(\cdot, \mathbf{x})$ is a smooth function on $\mathbb{R}^d$ for all $\mathbf{x} \in \mathcal{X}$;*

- $f(\boldsymbol{\theta}, \mathbf{x})$ is $\alpha$-homogeneous as in **C2**;

- the data is separated by $f$ during gradient descent at some point $t_0$;

- the learning rate satisfy $\eta_t := \eta_0 \lesssim \left( L(\boldsymbol{\theta}^{(t)}; \mathbf{w}) \log \left( 1/L(\boldsymbol{\theta}^{(t)}; \mathbf{w}) \right)^{3-2/\alpha} \right)^{-1}$ for all $t$,

then under exponential loss we have:

$$\frac{1}{L(\boldsymbol{\theta}^{(t)}; \mathbf{w})^2 \left( \log \frac{1}{L(\boldsymbol{\theta}^{(t)}; \mathbf{w})} \right)^{2-2/\alpha}} \geq \frac{1}{2} \alpha^2 \tilde{\gamma} \left( \boldsymbol{\theta}^{(t_0)}(\mathbf{w}) \right)^{2/\alpha} \sum_{i=t_0}^{(t)} \eta_i.$$

To use the results of Lemma A.1, we simply need to show two things for weak regularization:

- the total risk is still smooth and we still can achieve zero risk;

- there exists a critical (stationary) point such that $\lim_{\lambda \to 0} L_\lambda(\boldsymbol{\theta}^*; \mathbf{w}) = 0$.

Notice that the risk without regularization is a smooth function in terms of $\boldsymbol{\theta}$ for all $\mathbf{x}$, since the composition of smooth functions is still smooth. It is easy to see that adding a weak regularization, e.g. $\lambda \|\boldsymbol{\theta}\|_2^r$ for $r > 1$, does not alter the smoothness condition as $\lambda \to 0$. However, the weak $\ell_1$ regularization will make the total risk non-smooth, and therefore we have excluded it from our discussion.

For the second point, it is obvious that $\|\boldsymbol{\theta}\|_2 \to \infty$ is a critical point under exponential loss when $\lambda \to 0$. Recall that:

$$L_\lambda(\theta; \mathbf{w}) = \frac{1}{n} \sum_i w_i \exp\left( -y_i f\left( \boldsymbol{\theta}/\|\boldsymbol{\theta}\|_2, \mathbf{x}_i \right) \cdot \|\boldsymbol{\theta}\|_2 \right) + \lambda \|\boldsymbol{\theta}\|_2^r,$$

and

$$\nabla L_\lambda(\theta; \mathbf{w}) = \frac{1}{n} \sum_i -w_i \exp\left( -y_i f\left( \boldsymbol{\theta}/\|\boldsymbol{\theta}\|_2, \mathbf{x}_i \right) \cdot \|\boldsymbol{\theta}\|_2 \right) \cdot y_i \nabla f\left( \boldsymbol{\theta}, \mathbf{x}_i \right) + \lambda \nabla \|\boldsymbol{\theta}\|_2^r.$$

Therefore, for both the loss function and gradient, the main term decreases exponentially fast as $\|\boldsymbol{\theta}\|_2$ increases, while the remainder terms are only polynomial in $\|\boldsymbol{\theta}\|_2$, so we can always find a small enough $\lambda$ that satisfy: $\lim_{\lambda \to 0} \lim_{\|\boldsymbol{\theta}\| \to \infty} L_\lambda(\theta; \mathbf{w}) = 0$ and $\lim_{\lambda \to 0} \lim_{\|\boldsymbol{\theta}\| \to \infty} \nabla L_\lambda(\theta; \mathbf{w}) = 0$, in the same fashion as we show in the (A.1) below.

From a standard result of gradient descent on smooth function, which we summarize in Lemma A.2, gradient descent will always converge to a critical (stationary) point for the weighted ERM problem.

**Lemma A.2** (Lemma 10 of Soudry et al. (2018)). *Let $L_\lambda(\boldsymbol{\theta}; \mathbf{w})$ be a $\mathcal{B}(\mathbf{w})$-smooth non-negative objective. With a constant learning rate $\eta_0 \lesssim \mathcal{B}(\mathbf{w})^{-1}$, the gradient descent sequence satisfies:*

- $\lim_{t \to \infty} \sum_{i=1}^t \left\| \nabla L_\lambda(\boldsymbol{\theta}^{(t)}; \mathbf{w}) \right\| < \infty$;

- $\lim_{t \to \infty} \nabla L_\lambda(\boldsymbol{\theta}^{(t)}; \mathbf{w}) = 0$.

Now we need to show that under appropriate learning rate, which is specified in Lemma A.1, gradient descent converges to the stationary point that corresponds to the zero risk under weak regularization. Using the result from Lemma A.1, notice that if $L_\lambda(\boldsymbol{\theta}^{(t)}; \mathbf{w})$ does not decrease to 0, then the denominator $L_\lambda(\boldsymbol{\theta}^{(t)}; \mathbf{w})^2 \left( \log \frac{1}{L_\lambda(\boldsymbol{\theta}^{(t)}; \mathbf{w})} \right)^{2-2/\alpha}$ is bounded from below.

However, there exists a constant learning rate such that $\sum_{i=t_0}^t \eta_i \to \infty$ as $t \to \infty$, which leads to contradiction. Therefore, for weighted ERM with weak regularization, gradient descent converges to the stationary point where $L_\lambda(\boldsymbol{\theta}^{(t)}; \mathbf{w}) = 0$.

Finally, we show to make $L_\lambda(\boldsymbol{\theta}^{(t)}; \mathbf{w}) \to 0$, we must have $\|\boldsymbol{\theta}^{(t)}(\mathbf{w})\| \to \infty$. We show by contradiction. Suppose $\|\boldsymbol{\theta}^{(t)}; \mathbf{w})\|$ is bounded from above by some constant $C > 0$, for all $\lambda < \tilde{\lambda}$ that we

choose later. So the loss function for each sample $i$ is bounded below by a positive value that depends on $C$: $w_i \exp(-y_i f(\boldsymbol{\theta}^{(t)}, \mathbf{x})) \geq l(C) > 0$. Hence, let $K := \tilde{\lambda}^{-1/(r+1)}$, then

$$
\begin{aligned}
l(C) \leq L_\lambda(\boldsymbol{\theta}_\lambda(\mathbf{w}); \mathbf{w}) &\leq L_\lambda(K\boldsymbol{\theta}^*; \mathbf{w}) \\
&\leq M \exp\left(-\tilde{\lambda}^{-\alpha/(r+1)} \cdot \gamma^*\right) + \tilde{\lambda}^{1/(1+r)};
\end{aligned}
\tag{A.1}
$$

and it easy obvious that RHS$\rightarrow 0$ for a sufficiently small $\tilde{\lambda}$, which contradicts $l(C) > 0$. Hence, we have $\|\boldsymbol{\theta}^{(t)}(\mathbf{w})\| \to \infty$ for all all $\lambda < \tilde{\lambda}$, which completes the proof. □

## A.2 Supplementary material for Section 3

We provide the proofs for Proposition 1 and 2 in this part of the appendix.

### A.2.1 Proof for Proposition 1

*Proof.* We first characterize the $1/\log t$ rate using asymptotic arguments similar to that of Soudry et al. (2018). The key purpose here is to rigorously show that importance weighting plays a negligible role in the asymptotic regime. Let $\boldsymbol{\delta}(t)$ be the residual term at step $t$:

$$
\boldsymbol{\delta}(t, \mathbf{w}) := \boldsymbol{\theta}^{(t)}(\mathbf{w}) - \boldsymbol{\theta}^* \log t.
\tag{A.2}
$$

To show the $1/\log t$ rate, we simply need to prove that $\|\boldsymbol{\delta}(t, \mathbf{w})\|$ is bounded for any $\mathbf{w} \in [1/M, M]^n$. Notice that

$$
\|\boldsymbol{\delta}(t+1, \mathbf{w})\|^2 = \|\boldsymbol{\delta}(t+1, \mathbf{w}) - \boldsymbol{\delta}(t, \mathbf{w})\|^2 + 2\big(\boldsymbol{\delta}(t+1, \mathbf{w}) - \boldsymbol{\delta}(t, \mathbf{w})\big)^\mathsf{T} \boldsymbol{\delta}(t, \mathbf{w}) + \|\boldsymbol{\delta}(t, \mathbf{w})\|^2.
$$

For the first term, we have:

$$
\begin{aligned}
&\left\|\boldsymbol{\delta}(t+1, \mathbf{w}) - \boldsymbol{\delta}(t, \mathbf{w})\right\|^2 \\
&= \left\| -\eta \nabla L\big(\boldsymbol{\theta}^{(t)}(\mathbf{w}); \mathbf{w}\big) - \boldsymbol{\theta}^*\big(\log(t+1) - \log(t)\big)\right\|^2 \\
&= \eta^2 \left\| -\eta \nabla L\big(\boldsymbol{\theta}^{(t)}(\mathbf{w}); \mathbf{w}\big)\right\| + \|\boldsymbol{\theta}^*\|^2 \log^2(1 + 1/t) + 2\eta(\boldsymbol{\theta}^*)^\mathsf{T} \nabla L\big(\boldsymbol{\theta}^{(t)}(\mathbf{w}); \mathbf{w}\big) \log(1 + 1/t) \\
&\leq \eta^2 \left\| \nabla L\big(\boldsymbol{\theta}^{(t)}(\mathbf{w}); \mathbf{w}\big)\right\| + \|\boldsymbol{\theta}^*\|^2 t^{-2};
\end{aligned}
$$

where in the last line we use:

- $\forall u > 0, \log(1 + u) \leq u$;

- $(\boldsymbol{\theta}^*)^\mathsf{T} \nabla L\big(\boldsymbol{\theta}^{(t)}(\mathbf{w}); \mathbf{w}\big) = \sum_i -w_i \exp(-y_i \boldsymbol{\theta}^* \mathbf{x}_i) y_i \boldsymbol{\theta}^* \mathbf{x}_i \leq 0$ because $\boldsymbol{\theta}^*$ separates the data.

Also, from the first conclusion of Lemma A.2, we see that $\left\| \nabla L\big(\boldsymbol{\theta}^{(t)}(\mathbf{w}); \mathbf{w}\big)\right\| = o(1/t)$, so $\left\|\boldsymbol{\delta}(t+1, \mathbf{w}) - \boldsymbol{\delta}(t, \mathbf{w})\right\|^2 = o(1/t)$ and the running sum converges to some finite number:

$$
\sum_{t=1}^\infty \left\|\boldsymbol{\delta}(t+1, \mathbf{w}) - \boldsymbol{\delta}(t, \mathbf{w})\right\|^2 = C_0 < \infty.
$$

We see that the role of the weights is totally negligible because $\boldsymbol{\theta}^*$ separates the data (the second bullet point above). The same argument applies to the second term $2\big(\boldsymbol{\delta}(t+1, \mathbf{w}) - \boldsymbol{\delta}(t, \mathbf{w})\big)^\mathsf{T} \boldsymbol{\delta}(t, \mathbf{w})$, where $\mathbf{w}$ plays no part as long as $\boldsymbol{\theta}^*$ separates the data. The detailed proof is technical, and we refer to Lemma 6 of Soudry et al. (2018), which states that:

$$
\big(\boldsymbol{\delta}(t+1, \mathbf{w}) - \boldsymbol{\delta}(t, \mathbf{w})\big)^\mathsf{T} \boldsymbol{\delta}(t, \mathbf{w}) = o(1/t).
$$

Therefore, by applying tensorization, it holds that:

$$
\left\|\boldsymbol{\delta}(t, \mathbf{w})\right\|^2 - \left\|\boldsymbol{\delta}(t=0, \mathbf{w})\right\|^2 \leq C_0 + \sum_{i=1}^t \big(\boldsymbol{\delta}(t+1, \mathbf{w}) - \boldsymbol{\delta}(t, \mathbf{w})\big)^\mathsf{T} \boldsymbol{\delta}(t, \mathbf{w}) < \infty,
$$

hence $\|\boldsymbol{\delta}(t, \mathbf{w})\|$ is bounded and

$$\|\boldsymbol{\delta}(t, \mathbf{w})\| / \log t = \mathcal{O}(1/\log t), \quad \left| \frac{\boldsymbol{\theta}^{(t)}(\mathbf{w})}{\|\boldsymbol{\theta}^{(t)}(\mathbf{w})\|_2} - \boldsymbol{\theta}^* \right| = \mathcal{O}(\frac{1}{\log t}). \tag{A.3}$$

It is now obvious that under the asymptotic characterization of (A.2), the weights only play a negligible role since $\boldsymbol{\theta}^*$ separate the data. However, the definition of $\boldsymbol{\delta}$ under (A.2) also prohibits us from studying the finite-step behavior since it absorbs all the constant factors.

Now we use the Fenchel-Young inequality to give a more precise characterization of the convergence speed. First of all, recall the max-margin problem for linear predictor has a dual representation for separable data according to the KKT condition for separable problem:

$$\boldsymbol{\theta}^* = y_i \mathbf{X}_i \cdot p_i^* / \gamma^*, \tag{A.4}$$

where $p_i^*$ is the dual optimal such that

$$\gamma^* = -\min \left\{ \max_i -y_i \mathbf{x}_i^\mathsf{T} \boldsymbol{\theta} \text{ s.t. } \|\boldsymbol{\theta}\| = 1 \right\} \equiv \min \left\{ \|y_i \mathbf{X}_i \cdot p_i\| \text{ s.t. } p_i \geq 0, \sum_i p_i = 1 \right\}.$$

Now, we directly work with $\left| \frac{\boldsymbol{\theta}^{(t)}(\mathbf{w})}{\|\boldsymbol{\theta}^{(t)}(\mathbf{w})\|_2} - \boldsymbol{\theta}^* \right|$:

$$\left| \frac{\boldsymbol{\theta}^{(t)}(\mathbf{w})}{\|\boldsymbol{\theta}^{(t)}(\mathbf{w})\|_2} - \boldsymbol{\theta}^* \right|^2 = 2 - \frac{2\langle \boldsymbol{\theta}^*, \boldsymbol{\theta}^{(t)}(\mathbf{w}) \rangle}{\|\boldsymbol{\theta}^{(t)}(\mathbf{w})\|_2},$$

and from (A.4) and Fenchel-Young inequality we have:

$$-\frac{\langle \boldsymbol{\theta}^*, \boldsymbol{\theta}^{(t)}(\mathbf{w}) \rangle}{\|\boldsymbol{\theta}^{(t)}(\mathbf{w})\|_2} = \frac{\langle \boldsymbol{p}^*, -y_i \mathbf{x}_i^{(\mathsf{T})} \boldsymbol{\theta}^{(t)}(\mathbf{w}) \rangle}{\gamma^* \|\boldsymbol{\theta}^{(t)}(\mathbf{w})\|_2} \leq \frac{g^*(\boldsymbol{p}^*) + g\big(-y_i \mathbf{x}_i^{(\mathsf{T})} \boldsymbol{\theta}^{(t)}(\mathbf{w})\big)}{\gamma^* \|\boldsymbol{\theta}^{(t)}(\mathbf{w})\|_2}, \tag{A.5}$$

where $g$ is a convex function with it conjugate function given by $g^*$. To build the connections with the loss function and risk, we choose $g$ such that $g(\boldsymbol{u}) = \log \frac{1}{n} \sum_i w_i \exp(u_i)$. As a consequence, by letting $u_i = -y_i \mathbf{x}_i^{(\mathsf{T})} \boldsymbol{\theta}^{(t)}$ and $\boldsymbol{u} = [u_1, \ldots, u_n]$, we have $g(\boldsymbol{u}) = L(\boldsymbol{\theta}^{(t)}; \mathbf{w})$.

With simple algebraic computations, the conjugate function $g^*(\boldsymbol{p})$ is given by:

$$g^*(\boldsymbol{p}) = \log n + \sum_i p_i \log \frac{p_i}{w_i} = D_{KL}(\boldsymbol{p}\|\mathbf{w}) + \log n.$$

Plugging the above results to (A.5):

$$\frac{1}{2} \left| \frac{\boldsymbol{\theta}^{(t)}(\mathbf{w})}{\|\boldsymbol{\theta}^{(t)}(\mathbf{w})\|_2} - \boldsymbol{\theta}^* \right|^2 \leq 1 + \frac{\log L(\boldsymbol{\theta}^{(t)}(\mathbf{w}); \mathbf{w})}{\|\boldsymbol{\theta}^{(t)}(\mathbf{w})\|_2 \gamma^*} + \frac{\log n + D_{KL}(\boldsymbol{p}\|\mathbf{w})}{\|\boldsymbol{\theta}^{(t)}(\mathbf{w})\|_2 \gamma^*} \tag{A.6}$$

According the convergence analysis of Adaboost, we have the following technical lemma.

**Lemma A.3** (Schapire & Freund (2013)). *Suppose $\ell$ is convex, $\ell' \leq \ell$, and $\ell'' \leq \ell$, with a linear predictor and a sufficiently small learning rate such that $\eta_t L(\boldsymbol{\theta}^{(t)}) \leq 1$, then:*

$$L(\boldsymbol{\theta}^{(t+1)}) \leq L(\boldsymbol{\theta}^{(t)}) \Big( 1 - \eta_t L(\boldsymbol{\theta}^{(t)}) \big( 1 - \eta_t L(\boldsymbol{\theta}^{(t)})/2 \big) \Big( \frac{\|\nabla L(\boldsymbol{\theta}^{(t)})\|_2}{L(\boldsymbol{\theta}^{(t)})} \Big)^2 \Big), \tag{A.7}$$

*and thus*

$$L(\boldsymbol{\theta}^{(t+1)}) \leq L(\boldsymbol{\theta}^{(0)}) \exp \Big( -\sum_{j<t} \eta_t L(\boldsymbol{\theta}^{(j)}) \big( 1 - \eta_j L(\boldsymbol{\theta}^{(j)})/2 \big) \Big( \frac{\|\nabla L(\boldsymbol{\theta}^{(j)})\|_2}{L(\boldsymbol{\theta}^{(j)})} \Big)^2 \Big). \tag{A.8}$$

*Also, $\|\boldsymbol{\theta}^{(t+1)}\| \leq \sum_{j<t} \eta_t L(\boldsymbol{\theta}^{(j)}) \frac{\|\nabla L(\boldsymbol{\theta}^{(j)})\|_2}{L(\boldsymbol{\theta}^{(j)})}$.*

To use the results in Lemma A.3, we define the following shorthand notations. Let $a_t(\mathbf{w}) := \eta_t L(\boldsymbol{\theta}^{(t)}; \mathbf{w})$ and $b_t(\mathbf{w}) := \dfrac{\|\nabla L(\boldsymbol{\theta}^{(t)}(\mathbf{w}); \mathbf{w})\|_2}{L(\boldsymbol{\theta}^{(t)}(\mathbf{w}); \mathbf{w})}$. Now, (A.6) can be further given by:

$$
\frac{1}{2} \left| \frac{\boldsymbol{\theta}^{(t)}(\mathbf{w})}{\left\|\boldsymbol{\theta}^{(t)}(\mathbf{w})\right\|_2} - \boldsymbol{\theta}^* \right|^2 \leq 1 + \frac{\log L(\boldsymbol{\theta}^{(0)}; \mathbf{w})}{\|\boldsymbol{\theta}^{(t)}\|\gamma^*} -
$$
$$
\frac{\sum_{i=0}^{t-1} a_i(\mathbf{w})(1 - a_i(\mathbf{w})/2) b_i(\mathbf{w})^2}{\|\boldsymbol{\theta}^{(i)}\|\gamma^*} + \frac{\log n + D_{KL}(\boldsymbol{p}\|\mathbf{w})}{\left\|\boldsymbol{\theta}^{(t)}(\mathbf{w})\right\|_2 \gamma^*}
$$
$$
\leq 1 - \frac{\sum_{i=1}^{t-1} a_i(\mathbf{w}) b_i^2(\mathbf{w})}{\|\boldsymbol{\theta}^{(i)}\|\gamma^*} + \frac{2 \sum_{i=1}^{t-1} a_i^2(\mathbf{w}) b_i^2(\mathbf{w})}{\|\boldsymbol{\theta}^{(i)}\|\gamma^*} + \frac{\log n + D_{KL}(\boldsymbol{p}\|\mathbf{w})}{\left\|\boldsymbol{\theta}^{(t)}(\mathbf{w})\right\|_2 \gamma^*}.
$$
$$
\text{(A.9)}
$$

Notice that Lemma A.3 also imply:

$$
\sum_{i=1}^{t-1} a_i^2(\mathbf{w}) b_i^2(\mathbf{w}) = \sum_{i=1}^{t-1} \eta_i \|\nabla L(\boldsymbol{\theta}^{(i)}(\mathbf{w}); \mathbf{w})\| \leq 2 \sum_{i=1}^{t-1} \Big( L(\boldsymbol{\theta}^{(i)}(\mathbf{w}); \mathbf{w}) - L(\boldsymbol{\theta}^{(i+1)}(\mathbf{w}); \mathbf{w}) \Big),
$$

which is bounded from above by $2M$. Finally, it is easy to verify that $b_t(\mathbf{w}) \geq \gamma^*$, and Lemma A.3 also implies that $\|\boldsymbol{\theta}^{(t)}(\mathbf{w})\| \leq \sum_{i<t} a_i(\mathbf{w}) b_i(\mathbf{w})$. Finally, we simplify (A.9) to:

$$
\left| \frac{\boldsymbol{\theta}^{(t)}(\mathbf{w})}{\left\|\boldsymbol{\theta}^{(t)}(\mathbf{w})\right\|_2} - \boldsymbol{\theta}^* \right|^2 \leq 2 \cdot \frac{\log n + D_{KL}(\boldsymbol{p}\|\mathbf{w}) + M}{\left\|\boldsymbol{\theta}^{(t)}(\mathbf{w})\right\|_2 \gamma^*},
$$

and obtain the desired result. □

### A.3 Proof for Proposition 2

We first present a greedy approach for the construction of the maximal separable subset $\mathcal{D}_{\text{sep}}$, which is proposed by Ji & Telgarsky (2018b).

For each sample $(\mathbf{x}_i, y_i)$, if there exists a $\boldsymbol{\theta}_i$ such that $y_i \boldsymbol{\theta}_i^\mathsf{T} \mathbf{x}_i > 0$ and $\min_{j=1,\dots,n} y_j \boldsymbol{\theta}_i^\mathsf{T} \mathbf{x}_j \geq 0$, we add it to $\mathcal{D}_{\text{sep}}$. Otherwise, we add it to $\mathcal{D}_{\text{non-sep}}$. To see why this approach work, first notice that by choosing $\boldsymbol{\theta}_{sep}^* = \sum_{i \in \mathcal{D}} \boldsymbol{\theta}_i$, $\boldsymbol{\theta}_{sep}^*$ separates the data in $\mathcal{D}_{\text{sep}}$. Then we check it is indeed maximal: for any $\boldsymbol{\theta}$ that is correct on any $(\mathbf{x}_i, y_i)$ in $\mathcal{D}_{\text{non-sep}}$, there must also exist another $(\mathbf{x}_j, y_j)$ in $\mathcal{D}_{\text{non-sep}}$ so $y_i \boldsymbol{\theta}_i^\mathsf{T} \mathbf{x}_i < 0$, or otherwise $(\mathbf{x}_i, y_i)$ would have been in $\mathcal{D}_{\text{sep}}$.

It has been shown in Ji & Telgarsky (2018b) that the risk is strongly convex on $\mathcal{D}_{\text{non-sep}}$ under conditions that are satisfied by our setting.

**Lemma A.4** (Theorem 2.1 of Ji & Telgarsky (2018b)). *If $\ell$ is twice differentiable, $\ell'' > 0$, $l \geq 0$ and $\lim_{u \to \infty} \ell(u) = 0$, then $L(\boldsymbol{\theta}) = \sum_i \frac{1}{n} \ell(y_i \boldsymbol{\theta}^\mathsf{T} \mathbf{x}_i)$ is strongly convex on $\mathcal{D}_{\text{non-sep}}$.*

Now we provide the proof for Proposition 2.

*Proof.* The first part is a direct consequence of Lemma A.4, that $L(\boldsymbol{\theta}; \mathbf{w}) = \frac{1}{n} \sum_i w_i \exp(-y_i \boldsymbol{\theta}^\mathsf{T} \mathbf{x}_i)$ is strongly convex on $\mathcal{D}_{\text{non-sep}}$. Therefore, the optimum $\tilde{\boldsymbol{\theta}}(\mathbf{w})$ is uniquely defined and $\|\tilde{\boldsymbol{\theta}}(\mathbf{w})\| = \mathcal{O}(1)$. To show the second part, we leverage a standard argument for gradient descent with smoothness condition.

**Lemma A.5** (Bubeck (2014)). *Suppose $L(\boldsymbol{\theta})$ is convex and $\beta$-smooth. Then with learning rate $\eta_t \leq \beta/2$, the sequence of gradient descent satisfies:*

$$
L(\boldsymbol{\theta}^{(t+1)}) \leq L(\boldsymbol{\theta}^{(t)}) - \eta_t \big(1 - \eta_t \beta/2\big) \|\boldsymbol{\theta}^{(t)})\|^2.
$$

*Then for any $\mathbf{z} \in \mathbb{R}^d$:*

$$
2 \sum_{i=0}^{t-1} \eta_i \big(L(\boldsymbol{\theta}^{(i)}) - L(\mathbf{z})\big) \leq \|\boldsymbol{\theta}^{(0)} - \mathbf{z}\|^2 - \|\boldsymbol{\theta}^{(t)} - \mathbf{z}\|^2 + \sum_{i=0}^{t-1} \frac{\eta_i}{1 - \beta\eta_i/2} \big(L(\boldsymbol{\theta}^{(i)}) - L(\mathbf{z})\big).
$$

It is immediately clear that we may choose the $\mathbf{z}$ in Lemma A.5 such that it combines the optimal from $\mathcal{D}_{\text{sep}}$ and $\mathcal{D}_{\text{non-sep}}$. In particular, we have shown that the optimal on $\mathcal{D}_{\text{non-sep}}$ is uniquely given by $\tilde{\boldsymbol{\theta}}(\mathbf{w})$. For $\mathcal{D}_{\text{sep}}$ we assume the max-margin linear predictor is given by $\boldsymbol{\theta}_{sep}^*$ (so $\|\boldsymbol{\theta}_{sep}^*\| = 1$). Therefore, according to Proposition 1, the optimum is given by $\log t \cdot \boldsymbol{\theta}_{sep}^*$.

Now define

$$\mathbf{z} := \tilde{\boldsymbol{\theta}}(\mathbf{w}) + \boldsymbol{\theta}_{sep}^* \cdot \log t / \gamma_{sep},$$

where we add the extra constant $\gamma_{sep}$, which is the maximum margin on the separable subset of the data, to simplify the following bound. Without loss of generality, we assume the features are bounded in $\| \cdot \|_2$ norm such that $\|\mathbf{x}_i\|_2 \leq 1$. As a consequence:

$$L(\boldsymbol{\theta}; \mathbf{w}) = L_{\text{non-sep}}(\tilde{\boldsymbol{\theta}}(\mathbf{w}); \mathbf{w}) + L_{\text{sep}}(\mathbf{z}) \leq \inf_{\boldsymbol{\theta}} L(\boldsymbol{\theta}; \mathbf{w}) + n \exp(\|\tilde{\boldsymbol{\theta}}(\mathbf{w})\|)/t, \tag{A.10}$$

where we use $L_{\text{non-sep}}$ and $L_{\text{sep}}$ to denote the risk associated with $\mathcal{D}_{\text{non-sep}}$ and $\mathcal{D}_{\text{sep}}$. To invoke Lemma A.5, first note that the required smoothness condition is guaranteed by Lemma A.3, i.e. in each step, the risk is $\eta_t L(\boldsymbol{\theta}^{(t)})$-smooth. Without loss of generality, we assume $\eta_t L(\boldsymbol{\theta}^{(t)}) \leq \eta_t$. Therefore, according to Lemma A.5, we have:

$$
\begin{aligned}
2\Big(\sum_{i<t} &\eta_j\Big)\big(L(\boldsymbol{\theta}^{(i)}; \mathbf{w}) - L(\mathbf{z}; \mathbf{w})\big) \\
&\leq 2\sum_{i<t} \eta_j\big(L(\boldsymbol{\theta}^{(i)}; \mathbf{w}) - L(\mathbf{z}; \mathbf{w})\big) + 2\big(L(\boldsymbol{\theta}^{(i+1)}; \mathbf{w}) - L(\boldsymbol{\theta}^{(i)}; \mathbf{w})\big) \\
&\leq 2\sum_{i<t} \eta_j\big(L(\boldsymbol{\theta}^{(i)}; \mathbf{w}) - L(\mathbf{z}; \mathbf{w})\big) - \sum_{i<t} \frac{\eta_i}{1 - \eta_i/2}\big(L(\boldsymbol{\theta}^{(i)}; \mathbf{w}) - L(\boldsymbol{\theta}^{(i+1)}; \mathbf{w})\big) \\
&\leq \|\boldsymbol{\theta}^{(0)} - \mathbf{z}\|^2 - \|\boldsymbol{\theta}^{(t)} - \mathbf{z}\|^2 \leq \|\mathbf{z}\|^2.
\end{aligned}
\tag{A.11}
$$

Therefore, by our choice of $\mathbf{z}$ as well as the result in (A.10), we obtain the bound in terms of the risk:

$$L(\boldsymbol{\theta}^{(t)}; \mathbf{w}) \leq \inf_{\boldsymbol{\theta}} L(\boldsymbol{\theta}; \mathbf{w}) + \frac{\exp(\tilde{\boldsymbol{\theta}}(\mathbf{w}))}{t} + \frac{\|\tilde{\boldsymbol{\theta}}(\mathbf{w})\|^2 + \log^2 t/\gamma_{\text{sep}}^2}{2\sum_{i<t} \eta_i}.$$

Since we assume a constant learning rate, when $\sum_{i<t} \eta_i = \mathcal{O}(t)$ we can simplify the above result to:

$$L(\boldsymbol{\theta}^{(t)}; \mathbf{w}) \leq \inf_{\boldsymbol{\theta}} L(\boldsymbol{\theta}; \mathbf{w}) + \frac{C\big(\|\tilde{\boldsymbol{\theta}}(\mathbf{w})\|\big) + \log^2 t/\gamma_{\text{sep}}^2}{t}.$$

Finally, from Lemma A.4 we known $L(\boldsymbol{\theta}; \mathbf{w})$ is strongly convex (which we assume to be $\omega$-strongly-convex). So the convergence in terms of the risk can be transformed to parameters:

$$
\begin{aligned}
\big|\Pi_{\text{non-sep}}\boldsymbol{\theta}^{(t)}(\mathbf{w}) - \tilde{\boldsymbol{\theta}}(\mathbf{w})\big| &\leq \frac{2}{\omega}\Big(L_{\text{non-sep}}(\boldsymbol{\theta}^{(t)}(\mathbf{w}); \mathbf{w}) - L_{\text{non-sep}}(\tilde{\boldsymbol{\theta}}(\mathbf{w}); \mathbf{w})\Big) \\
&\leq \frac{2}{\omega}\Big(L(\boldsymbol{\theta}^{(t)}(\mathbf{w}); \mathbf{w}) - \inf_{\boldsymbol{\theta}} L(\boldsymbol{\theta}; \mathbf{w})\Big),
\end{aligned}
$$

which leads to our desired results. $\qquad\square$

## A.4 SUPPLEMENTARY MATERIAL FOR SECTION 4

In this section, we establish the detailed proofs of Proposition 3 and Theorem 1. Recall that the loss function we are interested in is:

$$\min_{\boldsymbol{\theta}} L_\lambda(\boldsymbol{\theta}; \mathbf{w}) := L(\boldsymbol{\theta}, \mathbf{w}) + \lambda\|\boldsymbol{\theta}\|^r, \tag{A.12}$$

Denote $\boldsymbol{\theta}_\lambda(\mathbf{w}) \in \arg\min L_\lambda(\boldsymbol{\theta}, \mathbf{w})$, $\theta^* = \arg\max_{\boldsymbol{\theta}:\|\boldsymbol{\theta}\|\leq 1} \max_i y_i f(\boldsymbol{\theta}, \mathbf{x}_i))$. Let $\gamma_\lambda(\mathbf{w}) = \max_i y_i f(\boldsymbol{\theta}_\lambda(\mathbf{w})/\|\boldsymbol{\theta}_\lambda(\mathbf{w})\|, \mathbf{x}_i)$, $\gamma^* = \max_i y_i f(\boldsymbol{\theta}^*, \mathbf{x}_i)$.

### A.4.1 PROOF OF PROPOSITION 3.

We first restate the proposition.

**Proposition A.1.** *Suppose C1, C2, A1 hold. For any* $\mathbf{w} \in [1/M, M]^n$*, it follows that*

- **(Asymptotic)** $\lim_{\lambda \to 0} \gamma_\lambda(\mathbf{w}) \to \gamma^*$.

- **(Finite steps)** *There exists a $\lambda := \lambda(r, \alpha, \gamma^*, \mathbf{w}, c)$ such that for $\boldsymbol{\theta}'(\mathbf{w})$ with $L_\lambda(\boldsymbol{\theta}'(\mathbf{w}); \mathbf{w}) \le \tau L_\lambda(\boldsymbol{\theta}_\lambda(\mathbf{w}); \mathbf{w})$ and $\tau \le 2$, the associated margin $\tilde{\gamma}(\boldsymbol{\theta}'(\mathbf{w}))$ satisfies $\tilde{\gamma}(\boldsymbol{\theta}'(\mathbf{w})) \ge c \cdot \frac{\gamma^*}{\tau^{\alpha/r}}$, where $\frac{1}{10} \le c < 1$*

**Proof of the Asymptotic part**:

*Proof.* We first take consider the exponential loss $\ell(u) = \exp(-u)$. The log loss $\ell(u) = \log(1 + \exp(-u))$ can be shown in a similar fashion. Suppose the weights $\mathbf{w} = (w_1, \ldots w_n)$ are normalized so that $\sum_{i=1}^n w_i = 1$ and $w_i \ge 0$. Consider

$$
\begin{aligned}
L_\lambda(A\boldsymbol{\theta}; \mathbf{w}) &= \sum_{i=1}^n w_i \exp(-A^\alpha \cdot y_i f(\boldsymbol{\theta}; \mathbf{x}_i)) + \lambda A^r \|\boldsymbol{\theta}\|^r \\
&\le \exp(-A^\alpha \cdot \max_i(y_i f(\boldsymbol{\theta}; \mathbf{x}_i))) + \lambda A^r \|\boldsymbol{\theta}\|^r,
\end{aligned}
\tag{A.13}
$$

where $A > 0$, and we disregard the $1/n$ term in $L_\lambda$ for the sake of notation. In addition, we have the lower bound

$$
\begin{aligned}
L_\lambda(A\boldsymbol{\theta}; \mathbf{w}) &\ge w_{i'} \cdot \exp(-A^\alpha \cdot \max_i(y_i f(\boldsymbol{\theta}; \mathbf{x}_i))) + \lambda A^r \|\boldsymbol{\theta}\|^r \\
&\ge w_{[n]} \cdot \exp(-A^\alpha \cdot \max_i(y_i f(\boldsymbol{\theta}; \mathbf{x}_i))) + \lambda A^r \|\boldsymbol{\theta}\|^r,
\end{aligned}
\tag{A.14}
$$

where $i' = \arg\min_i y_i f(\boldsymbol{\theta}; \mathbf{x}_i))$, $w_{[n]} = \min_i w_i$. By taking $A = \|\boldsymbol{\theta}_\lambda(\mathbf{w})\|$, $\boldsymbol{\theta} = \boldsymbol{\theta}^*$ in the upper bound and $A = 1$, $\boldsymbol{\theta} = \boldsymbol{\theta}_\lambda(\mathbf{w})$ in the lower bound , it follows that

$$
\begin{aligned}
& w_{[n]} \cdot \exp(-\|\boldsymbol{\theta}_\lambda(\mathbf{w})\|^\alpha \gamma_\lambda(\mathbf{w})) + \lambda \|\boldsymbol{\theta}_\lambda(\mathbf{w})\|^r \\
&\le L_\lambda(\mathbf{w})(\boldsymbol{\theta}_\lambda(\mathbf{w})) \\
&\le L_\lambda(\mathbf{w})(\|\boldsymbol{\theta}_\lambda(\mathbf{w})\|\boldsymbol{\theta}^*) \\
&\le \exp(-\|\boldsymbol{\theta}_\lambda(\mathbf{w})\|^\alpha \cdot \gamma^*) + \lambda \|\boldsymbol{\theta}_\lambda(\mathbf{w})\|^r.
\end{aligned}
$$

It implies that

$$
w_{[n]} \cdot \exp(-\|\boldsymbol{\theta}_\lambda(\mathbf{w})\|^\alpha \gamma_\lambda(\mathbf{w})) \le \exp(-\|\boldsymbol{\theta}_\lambda(\mathbf{w})\|^\alpha \cdot \gamma^*),
$$

or

$$
w_{[n]} \cdot \exp(-\|\boldsymbol{\theta}_\lambda(\mathbf{w})\|^\alpha (\gamma^* - \gamma_\lambda(\mathbf{w}))) \le 1.
$$

By Claim 1 that $\|\boldsymbol{\theta}_\lambda(\mathbf{w})\| \to \infty$ as $\lambda \to 0$ (or Lemma C.4 in Wei et al. (2019)), the above inequality implies that $\gamma_\lambda(\mathbf{w}) \to \gamma^*$ as $\lambda \to 0$. $\qquad\square$

**Proof of the Finite steps part**

*Proof.* Consider $A = [\frac{1}{\gamma^*} \log((\gamma^*)^{r/\alpha}/\lambda)]^{1/\alpha}$, it follows that

$$
\begin{aligned}
L_\lambda(\boldsymbol{\theta}'(\mathbf{w}), \mathbf{w}) &\le \tau L_\lambda(A\boldsymbol{\theta}^*) \\
&\le \tau \exp(-A^\alpha \cdot \gamma^*) + \tau \lambda A^r \qquad \text{[Upper Bound A.4.1]} \\
&= \frac{\lambda \tau}{(\gamma^*)^{r/\alpha}} \left(1 + (\log((\gamma^*)^{r/\alpha}/\lambda))^{r/\alpha}\right)
\end{aligned}
\tag{A.15}
$$

Then by the lower bound A.4.1, it follows that

$$
w_{[n]} \cdot \exp(-\|\boldsymbol{\theta}'(\mathbf{w})\|^\alpha \gamma'(\mathbf{w})) \le L_\lambda(\boldsymbol{\theta}'(\mathbf{w}), \mathbf{w}) \le A.15,
$$

where $\gamma'(\mathbf{w}) = \max_i y_i f(\mathbf{w}'/\|\mathbf{w}'\|, \mathbf{x}_i)$. Note $\lambda \|\boldsymbol{\theta}'(\mathbf{w})\|^r \le A.15$. It implies that

$$
\begin{aligned}
\gamma'(\mathbf{w}) &\ge \frac{-\log(A.15/w_{[n]})}{\|\boldsymbol{\theta}'(\mathbf{w})\|^\alpha} \\
&\ge \frac{-\log(\frac{\lambda \tau}{w_{[n]}(\gamma^*)^{r/\alpha}}(1 + (\log((\gamma^*)^{r/\alpha}/\lambda))^{r/\alpha}))}{\frac{\tau^{\alpha/r}}{\gamma^*}(1 + (\log((\gamma^*)^{r/\alpha}/\lambda))^{r/\alpha})^{\alpha/r}}
\end{aligned}
$$

Note that the numerator is at the scale $\log(\frac{1}{\lambda}/\log\frac{1}{\lambda})$ and the denominator is at the scale $\log\frac{1}{\lambda}$. So for sufficiently small $\lambda = \lambda(r, \alpha, \gamma^*, \mathbf{w}, c)$, we have $\gamma'(\mathbf{w}) \geq c \cdot \frac{\gamma^*}{\tau^{\alpha/r}}$, where $\frac{1}{10} \leq c < 1$. We leave the details of finding out the dependency of $\lambda(r, \alpha, \gamma^*, \mathbf{w}, c)$ on c to the readers, which is simply the basic analysis. $\qquad\square$

### A.4.2 PROOF OF THEOREM 1

When the training distribution $p_{\text{train}}$ deviates from the testing distribution $p_{\text{test}}$, we develop the generalization bound that characterizes this deviation. Denote by $p_s$ and $p_t$ the respective densities of $\mathbf{x}$ from the training data and the testing data. Let $D(P_t\|P_s) = \int \left((\frac{p_t(x)}{p_s(x)})^2 - 1\right)p_s(x)dx$ and $\eta(\mathbf{x}_i) = \frac{p_t(\mathbf{x}_i)}{p_s(\mathbf{x}_i)}$. We first restate Theorem 1:

**Theorem A.1.** *Assume $\sigma$ is 1-Lipschitz and 1-positive homogeneous. Then with probability at least $1 - \delta$, we have*

$$\mathbb{P}_{(\mathbf{x},y)\sim p_{\text{test}}}\Big(yf^{NN}(\boldsymbol{\theta}(\mathbf{w}),\mathbf{x}) \leq 0\Big) \leq$$

$$\underbrace{\frac{1}{n}\sum_{i=1}^{n}\eta(\mathbf{x}_i)\mathbf{I}\big(y_i f^{NN}(\boldsymbol{\theta}(\mathbf{w})/\|\boldsymbol{\theta}(\mathbf{w})\|,\mathbf{x}_i) < \gamma\big)}_{(I)} + \underbrace{\frac{C\cdot\sqrt{D(P_t\|P_s)+1}}{\gamma\cdot H^{(H-1)/2}\sqrt{n}}}_{(II)} + \epsilon(\gamma,n,\delta),$$

*where (I) is the empirical risk, (II) reflects the compounding effect of the model complexity of the class of $H$-layer neural networks and the deviation of the target distribution from the source distribution , $\epsilon(\gamma,n,\delta) = \sqrt{\frac{\log\log_2\frac{4C}{\gamma}}{n}} + \sqrt{\frac{\log(1/\delta)}{n}}$ is a small quantity compared to (I) and (II). Here $C := \sup_{\mathbf{x}}\|\mathbf{x}\|$; $\gamma$ is any positive value.*

To prove Theorem A.1, we first establish a few lemmas.

**Lemma A.6.** *Consider an arbitrary function class $\mathcal{F}$ such that $\forall f \in \mathcal{F}$ we have $\sum_{\mathbf{x}\in\mathcal{X}}|f(\mathbf{x})| \leq C$. Then, with probability at least $1 - \delta$ over the sample, for all margins $\gamma > 0$ and all $f \in \mathcal{F}$ we have,*

$$\mathbb{P}_{p_{(\mathbf{x},y)\sim p_{\text{test}}}}\Big(yf(\mathbf{x}) \leq 0\Big)$$

$$\leq \frac{1}{n}\sum_{i=1}^{n}\eta(\mathbf{x}_i)\mathbf{I}\big(y_i f(\mathbf{x}_i) < \gamma\big) + 4\frac{\mathcal{R}_{n,\boldsymbol{\eta}}(\mathcal{F})}{\gamma} + \sqrt{\frac{\log(\log_2\frac{4C}{\gamma})}{n}} + \sqrt{\frac{\log(1/\delta)}{2n}}, \qquad (A.16)$$

*where $\mathcal{R}_{n,\boldsymbol{\eta}}(\mathcal{F}) = \mathbb{E}\Big[\sup_{f\in\mathcal{F}}\frac{1}{n}\sum_{i=1}^{n}\eta(\mathbf{x}_i)f(\mathbf{x}_i)\epsilon_i\Big]$ is the weighted Rademacher complexity ($\epsilon_i$'s are i.i.d Rademacher variables).*

*Proof.* This lemma is adapted from Theorem 1 of Koltchinskii et al. (2002) by considering the deviation of the testing distribution from the training distribution. Then it is obtained following Theorem 5 of Kakade et al. (2009). $\qquad\square$

**Lemma A.7.** *Let $\mathcal{F}_H$ be the class of real-valued networks of depth $H$ over the domain $\mathcal{X}$, where each parameter matrix $W_h$ has Frobenius norm at most $M_F(h)$, and with an activation that is 1-Lipschitz, positive-homogeneous. Then,*

$$\mathcal{R}_{n,\boldsymbol{\eta}}(\mathcal{F}_H) \leq \frac{C\cdot\sqrt{D(P_t\|P_s)+1+o(\frac{1}{\sqrt{n}})}\cdot(\sqrt{2\log 2H}+1)}{\sqrt{n}}\prod_{h=1}^{H}M_F(h),$$

*where $C := \sup_{x\in\mathcal{X}}\|\mathbf{x}\|$.*

*Proof.* From Theorem 1 of Golowich et al. (2018), we arrive at

$$n\mathcal{R}(n,\boldsymbol{\eta})(\mathcal{F}_H) \leq \frac{1}{\lambda}\log\Big(2^H\cdot\mathbb{E}_{\boldsymbol{\epsilon}}\Big(M\lambda\|\sum_{i=1}^{n}\epsilon_i\eta(\mathbf{x}_i)\mathbf{x}_i\|\Big)\Big),$$

where $M = \prod_{h=1}^{H} M_F(h)$. Consider $Z := M \cdot \| \sum_{i=1}^{n} \epsilon_i \eta(\mathbf{x}_i) \mathbf{x}_i \|$ that is a random function of the $n$ Rademacher variables. Then

$$\frac{1}{\lambda} \log \left\{ 2^H \mathbb{E} \exp(\lambda Z) \right\} = \frac{H \log(2)}{\lambda} + \frac{1}{\lambda} \log \{ \mathbb{E} \exp \lambda (Z - \mathbb{E} Z) \} + \mathbb{E} Z.$$

By Jensen's inequality, we have

$$\mathbb{E}[Z] \le M \sqrt{\mathbb{E}_{\epsilon} \| \sum_{i=1}^{n} \epsilon_i \eta(\mathbf{x}_i) \mathbf{x}_i \|^2} = M \sqrt{\sum_{i=1}^{n} \eta(\mathbf{x}_i)^2 \| \mathbf{x}_i \|^2}.$$

In addition, we note that

$$Z(\epsilon_1, \ldots, \epsilon_i, \ldots, \epsilon_n) - Z(\epsilon_1, \ldots, -\epsilon_i, \ldots, \epsilon_n) \le 2 M \eta(\mathbf{x}_i) \| \mathbf{x}_i \|.$$

By the bounded-difference condition (Boucheron et al., 2013), $Z$ is a sub-Gaussian with variance factor $v = \frac{1}{4} \sum_{i=1}^{n} (2 M \eta(\mathbf{x}_i) \| \mathbf{x}_i \|)^2 = M^2 \sum_{i=1}^{n} \eta(x_i)^2 \| \mathbf{x}_i \|^2$. So

$$\frac{1}{\lambda} \{ \mathbb{E} \exp \lambda (Z - \mathbb{E} Z) \} \le \frac{\lambda M^2 \sum_{i=1}^{n} \eta(\mathbf{x}_i)^2 \| \mathbf{x}_i \|^2}{2}.$$

Taking $\lambda = \frac{\sqrt{2 \log(2) H}}{M \sqrt{\sum_{i=1}^{n} \eta(\mathbf{x}_i)^2 \| \mathbf{x}_i \|^2}}$, it follows that

$$\frac{1}{\lambda} \{ 2^H \mathbb{E} \exp \lambda Z \}$$

$$\le M (\sqrt{2 \log(2) H} + 1) \sqrt{\sum_{i=1}^{n} \eta(\mathbf{x}_i)^2 \| \mathbf{x}_i \|^2} \le \sqrt{n} C M (\sqrt{2 \log(2) H} + 1) \sqrt{\frac{1}{n} \sum_{i=1}^{n} \eta(\mathbf{x}_i)^2}.$$

$$\text{(A.17)}$$

By law of large number, $\frac{1}{n} \sum_{i=1}^{n} \eta(\mathbf{x}_i)^2 = D(P_t \| P_s) + 1 + o(\frac{1}{\sqrt{n}})$. The desired result follows. $\quad\square$

**Lemma A.8.** *Suppose $f^{NN}(\boldsymbol{\theta}, \cdot)$ is a $H$-layer neural network and $C = \sup_{x \in \mathcal{X}} \| x \|_2$. Then, There exists another parameter $\tilde{\boldsymbol{\theta}}$ s.t. $f^{NN}(\boldsymbol{\theta}/\| \boldsymbol{\theta} \|, \mathbf{x}) = f^{NN}(\tilde{\boldsymbol{\theta}}, \mathbf{x})$, for any $x \in \mathcal{X}$ and that*

- *the parameter matrix of each layer of $f^{NN}(\tilde{\boldsymbol{\theta}}, \cdot)$ has a Frobenius norm no larger than $1/\sqrt{H}$.*

- $\sup_{x \in \mathcal{X}} f^{NN}(\tilde{\boldsymbol{\theta}}, \cdot) \le C.$

*Proof.* This lemma are obtained by reorganizing the proof of Lemma D3 and the proof of Proposition D.1 of Wei et al. (2019). $\quad\square$

**Proof of Theorem A.1**

*Proof.* Theorem A.1 follows by Lemma A.6, A.7 and A.8. $\quad\square$

