# OpenReview forum: "Understanding the role of importance weighting for deep learning"
_ICLR.cc/2021/Conference — ICLR 2021 Spotlight_

### Official Review · AnonReviewer4 · 2020-10-22
**Interesting problem but the paper is not clea**

**Rating:** 7
**Confidence:** 4

**Review:**

It is now well-understood that when the data are linearly separable,  gradient descent over the linear class of functions converges toward the hard margin solution. It highlights the implicit bias of gradient descent. Among all solutions interpolating the dataset, gradient descent selects the one with larger margin, partly explaining why over-parametrized models may generalize.  The picture for non-linear class is a little bit more complicated.
In this paper, the authors study the impact of importance weighting on the implicit bias of gradient descent for both linear and non linear predictors. The problem is interesting and non trivial because importance weighting may affect the geometry of the gradient descent. In particular, a natural question is the following: does the importance weighting affect the margin of the solution (and thus change its generalization properties ?). Note that the authors does not clearly define what is the goal of the paper and to which problem there are trying to answer


Here are some questions and remarks about the paper:

- You study importance weighting for neural-networks: is it used in practice for these class of functions ? Could you give some references ?

- The introduction is not clear. For example the first paragraph aims to define importance sampling. Introduce a formal definition. You sentence "Importance weighting is a standard tool used to estimate a quantity under a target distribution while
only the source distribution is accessible" is not clear at all.  Then, you present exploratory ideas before clearly presenting the contributions of the paper. We are completely lost ... You should clearly define what is the goal of the paper and how you achieve it. I had to read 3 times before understanding what you really wanted to do

- Do you focus only on the binary case ? It is one again not clear "For the sake of notation, we mostly study the binary setting", if so just say it.

- You use concepts that you do not define clearly: what does mean that $\mathcal D$ is separated by $f(\theta^{t},x)$ at some point $x$ ? Do you mean that there exists $\theta^{t}$ such that  $y_if(\theta^{t},x_i) \geq 0$ ?

- The paragraph after claim 1 is not clear at all (and you forgot the exponent $\alpha$ on the norm of $\theta$). I really don't get what you are trying to say here ...

- There are problems with your bibliography (page 3 paragraph beginning section 3, page 12 paragraph beginning A.1.3 for examples)

- Sometimes you write $\|\cdot \|$ and sometimes $\|\cdot \|_2$. Keep the same notation along the paper


The proofs of the paper are a little bit hard to follow but seem to be correct. I also have an other question. Would you be able to generalize the analysis without the condition $w_i \in [1/M,M]$ ? This condition is restrictive and we cannot use $w_i = 0$ which can be very useful for robust purposes. So is it possible to look at the condition $w_i \leq M$ instead ? It is, in my opinion, an interesting question.


To summarize, the paper tackles an interesting problem (but in the restrictive setting $w_i \in [1/M,M]$)? However it paper is poorly written and not clear at all. I had the feeling that the authors were completely in a rush and did not organize the paper. Instead they put ideas together, without a clear common thread. I think you should focus on the form of the paper and then resubmit it for publication. There are too many problems right now. Work on the clarity of your ideas.

---

> ### Author Response · Authors · 2020-11-17
> **We hope that we address the concerns of the reviewer**
>
> We thank the reviewer for providing feedback to our manuscript. We apologize for the typos and clear writing, and we have corrected them in this version of the manuscript.
>
> We wish first to clarify that our research questions (Q1, Q2 and Q3) are defined clearly in Section 2. We believe that for the sake of readability to the general readers, it is necessary to first provide the background (on implicit bias, NN generalization, importance weighting, etc.) before stating our key research questions.
>
> We summarize our response to the individual remarks as below:
>
> Remark 1. "You study importance weighting for neural-networks: is it used in practice for these class of functions? Could you give some references ?"
>
> We kindly remind the reviewer that we have provided a number of references that use importance weighting for deep learning models, in the first two paragraphs in Section 1, and Section 5.
>
> Remark 2. "The introduction is not clear."
>
> We think the first sentence in the first graph of Section 1 is quite standard to introduce importance sampling, see [3].
>
> Our story line is as follows:
> a. Introduce the background of importance weighting, and demonstrate it is an active research topic.
> b. Introduce the existing importance weighting methods.
> c. Emphasize there lacks theoretical understanding of importance weighting:
>     Introduce the first result that studies the impact of importance weighting [3] from empirical observations;
>     Introduce implicit bias of GD, to which [3] connects the impact of importance weighting.
> d. State that we fill in the theoretical gap to understand importance weighting. Also provide a few directions that are worthy of further investigations.
>
> We think this story line is clear and we have already stated in the third and fourth paragraphs of Section 1 that our goal is to give a theoretical understanding of importance sampling in terms of the implicit bias of gradient descent as well as the generalization performance. The concrete research goals are defined as Q1, Q2, Q3 in Section 2.
>
> Remark 3. "Do you focus only on the binary case? It is one again not clear "For the sake of notation, we mostly study the binary setting", if so just say it."
>
> Extending our results to the multi-class case using the cross-entropy loss is straightforward consequence following the arguments in [1] and [2]. However, denoting the problem setup and providing visual illustration for multi-class classification require adding more notations that are unnecessary for our purpose. We do not think there is any overclaim in this case.
>
> Remark 4. "You use the concept "separable" that you do not define clearly."
>
> We kindly remind the reviewer that we define "separable" clearly in assumption A.1.
>
> Remark 5. "The paragraph after Claim 1 is not clear at all."
>
> The paragraph after Claim 1 is essential for the general audience to understand our work because it reveals the connection between norm divergence (Claim 1) and the max-margin solution. We believe our statement and message are clear.
>
> Remark 6. "There are problems with your bibliography"
>
> We thank the reviewer for pointing out this issue. They have been corrected.
>
> Remark 7. "The $\|\cdot\|_2$ norm notation is not consistent."
>
> We kindly remind the reviewer that we mentioned in the very beginning of the Preliminary section that we use $\|\cdot\|$ to denote the $\|\cdot\|_2$ norm when no confusion arises.
>
> Finally, we point out that our results do not depend on the lower bound of the weights, so technically speaking, we only require the weights to be bounded in [0,M]. However, when the importance weights are defined by the target and source distribution such as in the sample bias correction and domain adaptation, i.e. $w(x) = P_t(x) / P_s(x)$, the meaning of $w=0$ is unclear. Also, in the counterfactual machine learning where the importance weights are defined by the inverse of propensity score, i.e. $w(x)=1/\text{propensity}(x)$, having $w \in [1/M,M]$ is a common regularity condition. We decode to keep our conditions consistent with the above applications of importance weighting.
>
> We hope we have addressed the concerns of the reviewer, and again we thank the reviewer for the effort and time.
>
> [1]. Soudry, Daniel, et al. "The implicit bias of gradient descent on separable data." The Journal of Machine Learning Research 19.1 (2018): 2822-2878.
>
> [2]. Wei, Colin, et al. "Regularization matters: Generalization and optimization of neural nets vs their induced kernel." Advances in Neural Information Processing Systems. 2019.
>
> [3]. Jonathon Byrd and Zachary Lipton. What is the effect of importance weighting in deep learning? In
> International Conference on Machine Learning, pp. 872–881, 2019.

---

> > ### Comment · AnonReviewer4 · 2020-11-22
> > **Good answer from the reviewers**
> >
> > I closely looked at reviewer's comments and it is more clear now.
> > Consequently, I think the paper deserves to be accepted for publication and I change my score from 5 to 7.

---

### Official Review · AnonReviewer2 · 2020-10-28
**Novel results for weighted ERM and nonlinear models. The main messages of the paper can be improved.**

**Rating:** 7
**Confidence:** 4

**Review:**


### Summary

This paper studies the inductive bias of gradient descent (GD) on smooth non-linear models when optimizing a weighted ERM. The authors provide several novel results for the linear and non-linear model cases. For linear models and linearly separable data, they show that GD converges to the hard-margin SVM solution and the convergence rate upper bound is lower for weighted ERMs that have higher weight on low margin points. They further characterize the inductive bias for non-linearly separable data, on a unique non-linearly separable subspace defined by Ji and Telgarsky (2018). For nonlinear models they consider a weak regularization setup. They show that asymptotically GD converges to a max margin predictor, which is similar to the non-weighted ERM case. They prove a generalization bound for weighted ERM and together with experiments provide insights on the generalization performance of GD in this case.

### Reason for score

The paper provides several novel theoretical results in a practical setting. The proof techniques might be useful for analyzing non-linear models in other settings. Although the writing and main "take-home" messages of the paper can be improved, overall I recommend for acceptance.

### Pros
1.	Novel theoretical results in several challenging settings (linear models with non-separable data and non-linear models).
2.	Analysis in a practical setting of weighted ERM.
3.	Novel theoretical insights that are corroborated with experiments.


### Cons

1.	Although the authors provide several insights on weighted ERMs, it is difficult to understand the main takeaways from the analysis. Specifically, under which conditions is it better to use weighted ERM and not ERM. How can this be derived from the analysis?
2.	The discussion of the generalization results in Section 4 is not clear. The first result shows that asymptotically GD obtains the same margin in the case of non-weighted and weighted ERMS. Therefore, this *may* hint that they have the same generalization performance. In contrast, the discussion after Theorem 1 suggests that weighted ERM can have better generalization performance. This should be clarified. I think that Figure 2c and Figure 2d are very interesting because they show the following: for unbalanced data, GD on weighted ERM and ERM obtains roughly the same margin, but it has different generalization performance in these cases. I think that the authors should discuss this and it may strengthen the insights provided by the analysis.
3.	In the introduction, the contribution of the paper is not so clear. For example, "characterize the impact of importance weighting…" is not clear. "We propose several exploratory topics…" – this seems like a minor contribution. I think that the authors should summarize the theoretical results and say how they relate to the observations of Byrd et al. (2019).
4.	In Section 3, it is claimed that the exponential loss is used WLOG. Does this mean that the results also hold for other losses? This is not clear.
5.	It is claimed that Figure (b) shows better performance but this figure only shows the margin. Is the test loss also better?
6.	In some cases, it seems that the informal claims made by the authors are too strong. For example, in page 5 it is said that " we provide a complete understanding", but the inductive bias of GD is not shown on the separable space in the case of non-linearly separable data (only for the non-separable space). In page 5, it is said that theta_sep in the separable region does not depend on w, although this is not shown formally. In page 4, it is said that the pivotal role of the margin for generalization of NNs is well understood. However, we are currently far from understanding generalization in NNs and the cited papers provide only *upper* bounds on the generalization error and might be very loose.

---

> ### Author Response · Authors · 2020-11-17
> **We thank the reviewer for careful reading and providing valuable feedback**
>
> We want to thank the reviewer for the careful reading and providing valuable feedback, in particular for the comments which we believe will help us bring a better version of this paper.
> We summarize our response to the review's comments as below.
>
> Comment 1. "Under which condition is it better to use weighted ERM and not ERM. How can this be derived from the analysis?"
>
> There should be no doubt that the weighted ERM is not worse than ERM if appropriate weights are used. The goal of this work is to understand how importance weighting impacts ERM. To this end, we wish first to distinguish the two scenarios:
> (a). the weights are defined by the problem, e.g. the propensity-weighted learning and domain adaptation;
> (b). the weights are the artifact introduced to improve the optimization (training) and generalization performance, e.g. we give the corrupted samples smaller weights.
>
> For scenario (a), our results establish the optimization and generalization guarantee, in terms of to which local minima will the GD converge, and how that specific local minimum can generalize.
>
> For scenario (b), our results suggest that (using the linear classifier as an example):
> (1). When the data is separable, by selecting the weights according to the (inverse) margin, the convergence to the max-margin solution can be made faster.
> (2). When the data is not separable, for the separable part of the data, the arguments in (1) apply; and for the non-separable part, the weights uniquely determine the intercept (shift) of the final solution on the non-separable data.
> In the second case and the first case where the convergence is not reached by finite-step optimization, our arguments in Sec 4.2 characterize the tradeoffs in terms of the weights selection.
> Hence, the potential benefit of weighted ERM can depend how the weights are selected, e.g. if they can represent the hard-to-classify degree.
>
> Comment 2. "The discussion of the generalization results in Section 4 is not clear."
>
> Here, the generalization performance depends on both the margin on the training data (sampled from the source distribution) and the discrepancy between the source and target distribution. Therefore, Prop 3 does not account for the discrepancy part, so it cannot imply that weighted ERM and non-weighted ERM have the same generalization performance.  Also, Prop 3 shows the invariance of margin only after convergence in the asymptotic sense, while in practice, the finite-step optimization is more realistic. In the finite-step setting, the weights can affect the margin of the solution so the generalization can differ from the non-weighted outcome.
> We are also very interested in the connection between Fig 2c and 2d pointed out by the reviewer. According to our preliminary study, explaining the phenomenon thoroughly would require notions other than margin to characterize other aspects of the solution. Unfortunately, we do not find a good way to incorporate our findings to this paper because most of our arguments are related to margin. Hence we decide to pursue it as a new research topic.
>
> Comment 3. "In the introduction, the contribution of the paper is not so clear."
>
> We agree that the contributions can be summarized more clearly, and the relation to Byrd et al. (2019) can be strengthened. We will add them to the final version of the paper. In terms of the exploratory topics, we hope that they could connect to the audience from other domains and motivate follow up work.
>
> Comment 4. "In Section 3, it is claimed that the exponential loss is used WLOG. Does this mean that the results also hold for other losses?"
>
> Our results hold for all the loss functions that have the exponential-tail behavior we summarized in Appendix A.1.1. We understand it is not mandatory for the reviewer to check the appendix, so we point it out here that they include: exponential loss, log loss and cross-entropy loss.
>
> Comment 5. "It is claimed that Fig 2b shows better performance but this figure only shows the margin. Is the test loss also better?"
>
> In our experiments, the test loss is indeed better when the solution has a smaller margin. However, we do not expect the relationship to hold in the general cases, so we do not present the results in our manuscript to avoid confusion.
>
> Comment 6. "In some cases, it seems that the informal claims made by the authors are too strong."
>
> We agree that we only study the major scenarios and certain cases are not covered, so we change the statement of "complete understanding" in our paper to "in-depth understanding" to describe our work more precisely. As for the claim that ``$\theta_{sep}$ in the separable region does not depend on w'', we would like to argue that it is indeed a straightforward consequence of Prop 1. Finally, we thank the reviewer for pointing out that the matching lower bound for NN complexity is still under active research, so we change the statement to "the role of margin for NN generalization is being explored actively".

---

> > ### Comment · AnonReviewer2 · 2020-11-21
> > **Thanks for the clarifications**
> >
> > I have read the other reviews and responses. I recommend for acceptance and did not change the score.

---

### Official Review · AnonReviewer3 · 2020-10-28
**Theoretical and empirical explanations of the role of importance weighting for (deep) learning models**

**Rating:** 7
**Confidence:** 1

**Review:**

Summary:

This paper proposes a theoretical explanation of the role of importance weighting with regards to the implicit bias of gradient descent (convergence to the same direction as the maximum-margin solution) and the generalization ability of the model.

The authors first extend the norm divergence result to a general setting where weak regularization is used.

For linear predictor and separable data, they show that importance weighting affects the convergence speed during gradient descent but not the convergence result or the $1 / log(t)$  rate. In particular, they find an expression of the constant term resulting from the importance weighting. Thus, using an “inverse margin weighted” method, the authors are able to accelerate gradient descent in a finite-step optimization setting.

For linear predictor and non-separable data, the authors show that importance weighting uniquely defines the solution on the non-separable subset, which can be seen as an intercept/constant term. The weights only control how the constant shifts on the non-separable data subset.

For non-linear predictor, they show that the optimal margin is reached regardless of the importance weighting choice under the infinitesimal weak regularization setting. They also show that in a finite-sample setting, importance weighting affects the generalization bound via the empirical risk and a term depending on the model complexity and the deviation between target and source distributions. The impact of importance weighting on the generalization ability of the model is also shown empirically.

Overall, the authors show that importance weighting can affect how fast the model separates the data and how fast the model converges to the max-margin solution for linear and non-linear predictor in some settings. They also deduce that giving more weights to the hard-to-classify points, corresponding to the small-margin samples, when used in importance weighting, is very important for the acceleration of the optimization. The authors conjecture and show empirically that the results still holds if the importance weights are jointly learned with the model


Pros:
The paper seems to offer important theoretical results and empirical validations regarding the role of importance weighting on the implicit bias of gradient descent and the generalization ability of linear and non-linear models in some settings.
The paper is clear and well written. There is a good balance of theoretical findings and empirical validations.

Cons:
This paper does not seem to have any major weaknesses.


Minor Comments:
-	There seems to be a typo in the 3rd sentence of the 2nd section: “from the which” should probably be "from which"
-	There seems to be a typo in the 2nd paragraph of page 4: “to understand the role of importance weighting” ("of" is probably missing)
-	There seems to be a typo at the top of page 5: “is able to accelerate” ("to" is probably missing)

---

> ### Author Response · Authors · 2020-11-17
> **We thank the reviewer for careful reading and providing feedback to our manuscript**
>
> We want to thank the reviewer for the careful reading of our manuscript and providing the feedback!
> We apologize for the typos and unclear writings, and we have corrected them in this version of the paper.
>
> We have also made minor modifications to the manuscript to address the issues pointed out by the other reviewers.

---

### Official Review · AnonReviewer1 · 2020-10-30
**Useful insights into importance weighting**

**Rating:** 7
**Confidence:** 4

**Review:**

The paper studies the effect of importance weighting schemes in deep learning models. A deep learning setting is considered where the empirical loss of labeled training data is weighted with importance weights and regularization on the network parameters is also included in the objective, which is optimized with gradient descent.

Two main results are presented in the paper. The first result focuses on a linear prediction scheme, in which case the convergence is shown to be faster if the weights of the samples are matched with the inverse of their SVM margin. An extension of this result is also proposed for the case where the data is not linearly separable.

The other main result considers multiple-layer feedforward networks in a covariate shift setting. A generalization bound is presented, which relates the probability of error to the agreement between the importance weights and the deviation between the source and the target distributions. The interesting takeaway message from this analysis is that aligning the weights with the distribution deviation reduces the error.

Overall, the paper is clear and well written, and it provides some interesting insights into several common learning settings from the perspective of importance weighting.

---

> ### Author Response · Authors · 2020-11-17
> **We thank the reviewer for careful reading and providing feedback**
>
> We want to thank the reviewer for the careful reading of our manuscript and providing the feedback!
>
> We have made minor modifications to the manuscript to address the issues pointed out by the other reviewers.

---

### Decision · Program_Chairs · 2021-01-07
**Final Decision**

**Decision:**

Accept (Spotlight)

**Comment:**

The paper studies the effect of importance weighting schemes on the implicit bias of gradient descent in deep learning models. It provides several theoretical results which give important insights on the effect of the importance weighting scheme on the limit of the convergence, as well as convergence rates. Results are presented for linear separators and deep learning models. A covariate shift setting is also studied. The theoretical results are supported with empirical demonstrations, and also lead to useful insights regarding which weighting schemes are expected to be more helpful. They also explain some previously observed empirical phenomena.


Pros:
- New theoretical results which provide important insights on an important topic
- Empirical demonstrations which support the theoretical results


Cons:
- No significant issues.